# Cost-Sensitive Multi-Fidelity Bayesian Optimization

## Abstract

In this paper, we address the problem of *cost-sensitive multi-fidelity* Bayesian Optimization (BO) for efficient hyperparameter optimization (HPO). Specifically, we assume a scenario where users want to early-stop the BO when the performance improvement is not satisfactory with respect to the required computational cost. Motivated by this scenario, we introduce *utility*, which is a function describing the trade-off between cost and performance of BO and can be estimated from the user's preference data. This utility function, combined with our novel acquisition function and stopping criterion, allows us to dynamically choose for each BO step the best configuration that we expect to maximally improve the utility in future, and also automatically stop the BO around the maximum utility. Further, we improve the sample efficiency of existing learning curve (LC) extrapolation methods with transfer learning to develop a sensible surrogate function for multi-fidelity BO. We validate our algorithm on various LC datasets and found it outperform all the previous multi-fidelity BO and transfer-BO baselines we consider, achieving significantly better trade-off between cost and performance of BO.

## 1 Introduction

Hyperparameter optimization (HPO) (Bergstra & Bengio, 2012; Bergstra et al., 2011; Hutter et al., 2011; Snoek et al., 2012; Cowen-Rivers et al., 2022; Li et al., 2018; Franceschi et al., 2017) stands as a crucial challenge in the domain of deep learning, given its importance of achieving optimal empirical performance. Unfortunately, the field of HPO for deep learning remains relatively underexplored, with many practitioners resorting to simple trial-and-error methods (Bergstra & Bengio, 2012; Li et al., 2018). Moreover, traditional black-box Bayesian optimization (BO) approaches for HPO (Bergstra et al., 2011; Snoek et al., 2012; Cowen-Rivers et al., 2022) have limitations when applied to deep neural networks due to the impracticality of evaluating a vast number of hyperparameter configurations until convergence, each of which may take several days.

Recently, multi-fidelity (or gray-box) BO (Li et al., 2018; Falkner et al., 2018; Awad et al., 2021; Swersky et al., 2014; Wistuba et al., 2022; Arango et al., 2023; Kadra et al., 2023; Rakotoarison et al., 2024) has gained increasing attention to improve the sample efficiency of traditional black-box BO. Multi-fidelity BO makes use of lower fidelity information (e.g., validation accuracies at fewer training epoches) to predict and optimize the performances at higher or full fidelity (e.g., validation accuracies at the last training epoch). Unlike black-box BO, multi-fidelity BO dynamically selects hyperparameter configurations even before finishing a single training run, demonstrating its ability of finding better configurations in a more sample efficient manner than black-box BO.

However, one critical limitation of the conventional multi-fidelity BO frameworks is that they are not aware of the trade-off between the cost and performance of BO. For instance, given a limited amount of total credits, customers of cloud computing services (e.g., GCP, AWS, or Azure) may choose to heavily penalize the cost of BO relative to its performance, in order to conserve credits for other tasks. A similar scenario applies to users of task managers such as Slurm, who aim to optimize their allocated time within a computing instance. In those cases, users may want the BO process to focus on exploiting the current belief on good hyperparameter configurations than trying to explore new configurations, in order to efficiently consuming their limited resources. Yet, the existing multi-fidelity BO methods tend to over-explore because they usually assume a sufficiently large budget (e.g., total credits, or allocated time) for the BO and aim to obtain the best asymptotic

performance on a validation set, hence are not able to properly penalize the cost (Swersky et al., 2014; Kadra et al., 2023). We could lower the total BO budget and maximize the performance at that maximum budget, but in practice it is hard to specify the target budget in advance as it is difficult to accurately predict the trajectory of the future BO performances.

Therefore, in this paper we introduce a more sophisticated notion of cost-sensitivity for multi-fidelity BO. Specifically, we assume that it is easier to specify the trade-off between cost and performance of multi-fidelity BO, than to know the proper target budget in advance. We call this trade-off *utility*. This utility function describes users' own preferences about the trade-off and can be estimated from the user's preference data. It has higher values as cost decreases and performance increases, and vice versa (Fig. 1a). Some users may want to strongly penalize the amount of BO budgets spent, while others

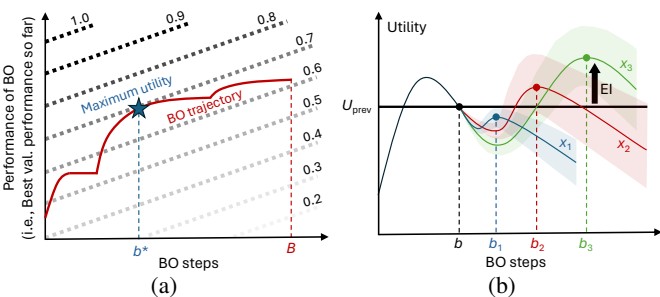

Figure 1: **(a)** A utility function shown in the dotted black lines. The red curve shows a BO trajectory from which we determine the maximum utility ($\approx 0.7$) and when to stop ($b^*$). **(b)** An illustration of selecting the best configuration at each BO step. Notice, the y-axis is utility. Starting from the current BO step $b$, we extrapolate the LCs with the three configurations $x_1, x_2, x_3$ (shown in the solid curves with colors and the shaded area), and then select $x_3$ which achieves at $b_3$ the maximum expected improvement (EI) of utility over the previous utility $U_{\text{prev}}$.

may weakly penalize or not penalize at all as with the conventional multi-fidelity BO. We explicitly maximize this utility by dynamically selecting hyperparameter configurations expected to achieve the greatest improvement in the future (Fig. 1b), and also automatically terminating the BO around the maximum utility (Fig. 1a), instead of terminating at an arbitrary target budget.

Solving this problem requires our multi-fidelity BO method to have the following capabilities. Firstly, it should support **freeze-thaw BO** (Swersky et al., 2014; Rakotoarison et al., 2024), an advanced form of multi-fidelity BO in which we can dynamically pause (freeze) and resume (thaw) hyperparameter configurations based on future performances extrapolated from a set of partially observed learning curves (LCs) with various configurations. Such efficient and sensible allocation of computational resources suits well for our purpose of finding the best trade-off between cost and performance of multi-fidelity BO. Secondly, freeze-thaw BO requires its surrogate function to be equipped with a **good LC extrapolation** mechanism (Adriaensen et al., 2023; Rakotoarison et al., 2024; Kadra et al., 2023). In our case, it is crucial for making a good probabilistic inference on future utilities with which we dynamically select the best configuration and accurately early-stop the BO. Lastly, since we assume that users want to stop the BO as early as possible, LC extrapolation should be accurate even at the very early stages of BO. Therefore, we should make use of **transfer learning** to maximally improve the sample efficiency of BO (Arango et al., 2023) and to prevent inaccurate early-stopping when there are only few or even no observations in the BO.

Based on those criteria, we introduce our novel **C**ost-sensitive **M**ulti-fidelity **BO** (CMBO) that can effectively maximize the utility based on the three components mentioned above. We first introduce the detailed notion of utility function and how to estimate it from the user's own preference data[1] (§3.1) . We then describe the acquisition function and stopping criterion specifically developed for our framework, and explain how to achieve a good trade-off between cost and performance of multi-fidelity BO with them (§3.2). Building on the recently introduced Prior-Fitted Networks (PFNs) (Müller et al., 2021; Adriaensen et al., 2023) for in-context Bayesian inference, we explain how to train a PFN with the existing LC datasets to develop a sample efficient in-context surrogate function for freeze-thaw BO that can also effectively capture the correlations between different hyperparameter configurations (§3.3). Lastly, we empirically demonstrate the superiority of CMBO on a set of diverse utility functions, three multi-fidelity HPO benchmarks, and one real-world object-detection LC dataset we collected, showing that it significantly outperforms all the previous multi-fidelity BO and the transfer-BO baselines we consider (§4).

---

[1]Some users may already have an exact form of their utility function, but for the others we need to provide a reasonable way to quantify it with their own preference data.

We summarize our contributions and findings as follows:

- We introduce the concept of utility, which describes the trade-off between cost and performance of multi-fidelity BO, along with the method to quantify it with user preference data.
- We propose a new problem formulation, cost-sensitive multi-fidelity HPO, where we aim to maximize utility instead of maximizing the asymptotic validation performances.
- We introduce our novel acquisition function and stopping criterion specifically designed for our problem formulation, along with the transfer learning of in-context LC extrapolation.
- We extensively validate the superiority of CMBO on various cost-sensitive multi-fidelity HPO settings, with three popular benchmarks and one real-world object detection LC dataset we collected.

## 2 RELATED WORK

We briefly discuss the related work in this section. See §A for the other related work on **multi-fidelity HPO**, **transfer BO**, **cost-sensitive HPO**, and **BO with user preference**.

**Freeze-thaw BO.** Freeze-thaw BO (Swersky et al., 2014) dynamically pauses (freezes) and resumes (thaws) configurations based on the last epoch performances extrapolated from a set of partially observed LCs obtained from other configurations, leading to an efficient and sensible allocation of computational resources. DyHPO (Wistuba et al., 2022) and its transfer version (Arango et al., 2023) improve the computational efficiency of freeze-thaw BO with deep kernel GP (Wilson et al., 2016), but their acquisition extrapolates the LCs only a one-step forward, producing a greedy strategy. Other recent variants of freeze-thaw BO include DPL (Kadra et al., 2023) and ifBO (Rakotoarison et al., 2024) which are not greedy, and their acquisitions maximize the performance either at the last BO step or random future steps. On the other hand, we maximize the utility specified by each user.

**Learning curve extrapolation.** Freeze-thaw BO requires the ability of dynamically updating predictions on future performances from partially observed LCs, thus heavily relies on LC extrapolation (Baker et al., 2017; Gargiani et al., 2019; Wistuba & Pedapati, 2020). DyHPO (Wistuba et al., 2022) and Quick-Tune (Arango et al., 2023) propose to extrapolate LCs for only a single step forward. Freeze-thaw BO (Swersky et al., 2014) and DPL (Kadra et al., 2023) use non-greedy extrapolations but limit the shape of the LCs. Domhan et al. (2015) consider a broader set of basis functions, but requires computationally expensive MCMC, and also do not consider correlations between different configurations. Klein et al. (2017b) models interactions between configurations with a Bayesian neural network (BNN), but suffers from the same computational inefficiency of MCMC and online retraining. LC-PFNs (Adriaensen et al., 2023) are an in-context Bayesian LC extrapolation method without retraining, but they do not consider interactions between configurations. Recently, ifBO (Rakotoarison et al., 2024) further combine LC-PFNs with PFNs4BO (Müller et al., 2023) to develop an in-context surrogate function for freeze-thaw BO, but they train PFNs only with a prior distribution. On the other hand, we use transfer learning, i.e., train PFNs with the existing LC datasets, to improve the sample efficiency of freeze-thaw BO while successfully encoding the correlations between configurations at the same time.

## 3 APPROACH

In this section, we introduce CMBO, a novel method for cost-sensitive multi-fidelity HPO. We first introduce notation, backgrounds on freeze-thaw BO, and utility function in §3.1. We then introduce the overall method and algorithm in §3.2, and the transfer learning of surrogate functions in §3.3.

### 3.1 BACKGROUNDS AND UTILITY FUNCTION

**Notation.** Following the convention, we assume that we are given a finite pool of hyperparameter configurations $\mathcal{X} = \{x_1, \ldots, x_N\}$, with $N$ the number of configurations. Let $t \in [T] := \{1, \ldots, T\}$ denote the training epochs, $T$ the last epoch, and $y_{n,1}, \ldots, y_{n,T}$ the validation performances (e.g., validation accuracies) obtained with the configuration $x_n$. We further introduce notations for multi-fidelity BO. Let $b = 1, \ldots, B$ denote the BO steps, $B$ the last BO step, and $\tilde{y}_1, \ldots, \tilde{y}_B$ the BO performances, i.e., each $\tilde{y}_b$ is the best validation performance ($y$) obtained until the BO step $b$.

**Freeze-thaw BO.** The goal of multi-fidelity BO is to find the optimal intermediate performance over the hyperparameter configurations, i.e., $\max_{n \in [N], t \in [T]} y_{n,t}$. Freeze-thaw BO (Swersky et al., 2014)

is an advanced form of multi-fidelity BO. At each BO step, it allows us to dynamically select and evaluate the best hyperparameter configuration $x_{n^*}$ with $n^* \in [N]$ denoting the corresponding index, while pausing the evaluation on the previous best configuration. Specifically, given $\mathcal{C} = \{(x, t, y)\}$ that represents a set of partial LCs collected up to a specific BO step, we predict for all $x \in \mathcal{X}$ the remaining part of the LCs up to the last training epoch $T$ with a (pretrained) LC extrapolator, compute the acquisition such as the expected improvement (Mockus et al., 1978) of validation performance at epoch $T$, and select and evaluate the best configuration $x_{n^*}$ that maximizes the acquisition. Note that at any BO step, the partial LCs in $\mathcal{C}$ can have different length across the configurations. Suppose that at BO step $b$ the next training epoch for $x_{n^*}$ is $t_{n^*}$. We then evaluate $x_{n^*}$ a single epoch from the corresponding checkpoint to obtain the validation performance $y_{n^*, t_{n^*}}$ at the next epoch $t_{n^*}$, which we use to update the corresponding partial LC in $\mathcal{C}$ and compute the BO performance $\tilde{y}_b$. We repeat this process $B$ times until convergence. See Alg. 1 for the pseudocode (except the red parts).

**Utility function.** A utility function $U$ describes the trade-off between the BO step $b$ and the BO performance $\tilde{y}_b$. Its values $U(b, \tilde{y}_b)$ negatively correlate with $b$ and positively with $\tilde{y}_b$. For instance, we may simply define $U(b, \tilde{y}_b) = \tilde{y}_b - \alpha b$ for some $\alpha > 0$, such that the utility gives linear incentives and penalties to the performance and number of BO steps, respectively. Or, we could use a weighted linear combination of linear, quadratic, and square root function, as appropriate (used to draw Fig. 2).

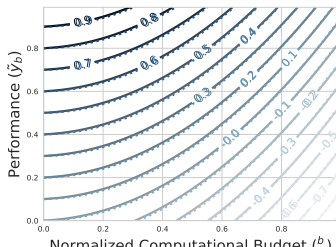

We assume that users have their own utility functions. However, it is often difficult for them to quantify the function. We thus propose to use Bradley-Terry model (Bradley & Terry, 1952):

Figure 2: An example of utility function estimation. True and estimated utility are denoted as black solid and blue dotted lines, respectively.

$$p(u_1 > u_2) = \frac{\exp(u_1/\tau)}{\exp(u_1/\tau) + \exp(u_2/\tau)}. \tag{1}$$

Eq. (1) is the probability that the user prefers $u_1$ to $u_2$, where $u_1$ and $u_2$ are the utility values of two different points in the space consisting of BO step $b$ and BO performance $\tilde{y}$, and $\tau$ is a temperature. Specifically, we first collect the **user preference data** by asking users to decide which points to prefer given a set of pair of points (Bai et al., 2022). We then optimize the parameters in the utility function (e.g., $\alpha$) by minimizing the binary cross entropy loss $-m \log p(u_1 > u_2) - (1 - m) \log(1 - p(u_1 > u_2))$ with gradient descent, where $m \in \{0, 1\}$ is the binary label of user preference.

Fig. 2 shows that the proposed method can recover the true utility function very accurately with 1,000 datapoints. See §B for the detailed experimental setups and other examples with fewer datapoints. In this way, for the rest of this paper, we assume that we can estimate the utility function with the user preference data, or users already know the exact form of their utility functions.

### 3.2 COST-SENSITIVE MULTI-FIDELITY BO

We next introduce our main method based on freeze-thaw BO and the notion of utility (§3.1).

**Acquisition function.** Let $t_n$ be the next training epoch for the configuration $x_n$ at a BO step $b$. Further, suppose we have a LC extrapolator $f(\cdot | x_n, \mathcal{C})$ that can probabilistically estimate $x_n$'s remaining part of LC, $y_{n, t_n:T}$, conditioned on $\mathcal{C}$ a set of partial LCs collected up to the step $b$. Then, based on the expected improvement (EI) (Mockus et al., 1978), we define the acquisition function $A$:

$$A(n) = \max_{\Delta t \in \{0, \ldots, T - t_n\}} \mathbb{E}_{y_{n, t_n:T} \sim f(\cdot | x_n, \mathcal{C})} \left[ \max \left( 0, U(b + \Delta t, \tilde{y}_{b + \Delta t}) - U_{\text{prev}} \right) \right]. \tag{2}$$

In Eq. (2), we first extrapolate $y_{n, t_n:T}$, the remaining part of the LC associated with $x_n$, and compute the corresponding predictive BO performances $\{\tilde{y}_{b + \Delta t} \mid \Delta t = 0, \ldots, T - t_n\}$. Note that according to the definition in §3.1, $\tilde{y}_{b + \Delta t}$ is computed by taking the maximum among the last step BO performance $\tilde{y}_{b-1}$ as well as the newly extrapolated validation performances $y_{n, t_n}, \ldots, y_{n, t_n + \Delta t}$. Then, based on the increased BO step $b + \Delta t$ and the updated BO performance $\tilde{y}_{b + \Delta t}$, we compute the corresponding utility, and its expected improvement over the previous utility $U_{\text{prev}}$ over the distribution of LC extrapolation with the Monte-Carlo (MC) estimation. The acquisition $A(n)$ for each configuration index $n$ is defined by picking the best increment $\Delta t \in \{0, \ldots, T - t_n\}$ that maximizes the expected improvement, and we eventually choose the best configuration index $n$ that maximizes $A$ (see Fig. 1b).

The main differences of our acquisition function in Eq. (2) from the EI-based acquisitions used in the previous works are twofold. First, instead of maximizing the expected improvement of validation performance $y$, we maximize the expected improvement of utility. Second, rather than setting the target epoch at which we evaluate the acquisition to the last epoch $T$, we dynamically choose the optimal target epoch that is expected to maximally improve the utility.

Those aspects allow our BO framework to more carefully select configurations for each BO step, seeking the best trade-off between cost and performance of BO. Specif-

---

**Algorithm 1** Cost-sensitive Multi-fidelity BO

1: **Input:** LC extrapolator $f$, acquisition function $A$, utility function $U$, maximum BO steps $B$, hyperparameter configuration pool $\mathcal{X}$, number of configurations $N$.
2: $U_{\text{prev}} \leftarrow 0$, $\tilde{y}_0 \leftarrow -\infty$, $\mathcal{C} \leftarrow \emptyset$, $t_1, \ldots, t_N \leftarrow 1$
3: **for** $b = 1, \ldots, B$ **do**
4:     $n^* \leftarrow \arg\max_n A(n)$   ▷ Acquisition func., Eq. (2)
5:     **if** Eq. (3) and $b > 1$ **then**   ▷ Stopping criterion
6:         Break the for loop   ▷ Stop the BO
7:     **end if**
8:     Evaluate $y_{n^*, t_{n^*}}$ with $x_{n^*}$.
9:     $\mathcal{C} \leftarrow \mathcal{C} \cup \{(x_{n^*}, t_{n^*}, y_{n^*, t_{n^*}})\}$ ▷ Update the history
10:    $\tilde{y}_b \leftarrow \max(\tilde{y}_{b-1}, y_{n^*, t_{n^*}})$   ▷ Update the BO perf.
11:    $U_{\text{prev}} \leftarrow U(b, \tilde{y}_b)$   ▷ Update the prev. utility
12:    $t_{n^*} \leftarrow t_{n^*} + 1$
13: **end for**

---

ically, the acquisition function initially prefers configurations that are expected to produce good asymptotic validation performances, but as the BO proceeds it will gradually become greedy as the performance of BO saturates and the associated cost dominates the utility function. As a result, the acquisition function will tend to exploit more than explore – it will try to avoid selecting new configurations but stick to the few current configurations to maximize the short term performances.

Note that $U_{\text{prev}}$ in Eq. (2), the threshold of EI, is *not* the greatest utility achieved so far, but simply set to the utility value achieved most recently (line 11 in Alg. 1). This is because the cost of BO that has previously been incurred is not reversible. It differs from the typical EI-based BO settings where all the previous evaluations are meaningful and we can set the threshold to the maximum among them. As a result, $U_{\text{prev}}$ can either increase or decrease during the BO, and we need to stop the BO when $U_{\text{prev}}$ starts decreasing monotonically, i.e., when the performance of BO stops improving meaningfully with respect to the associated cost.

**Stopping criterion.** The next question is how to properly stop the BO around the maximum utility. We propose to stop when the following criterion is satisfied at each BO step $b > 1$:

$$\frac{\hat{U}_{\max} - U_{\text{prev}}}{\hat{U}_{\max} - \hat{U}_{\min}} > \delta_b. \quad (3)$$

In Eq. (3), $U_{\text{prev}}$ is the utility value at the last step $b - 1$, $\hat{U}_{\max}$ is defined as the maximum utility value seen up to the last step, and $\hat{U}_{\min} = U(B, \tilde{y}_1)$. The role of $\hat{U}_{\max}$ and $\hat{U}_{\min}$ is to roughly estimate the maximum and the minimum utility achievable over the course of BO, respectively. Therefore, the LHS of Eq. (3) can be seen as the *normalized regret* of utility roughly estimated at the current step $b$, and we stop the BO as soon as the current estimation on the regret exceeds some threshold $\delta_b$.

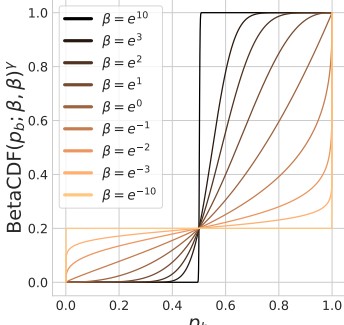

Figure 3: Eq. (4) with $\gamma = \log_2 5$ and the various values for $\beta$.

To define $\delta_b$, let $n^* = \arg\max_n A(n)$ denote the index of the currently chosen best configuration $x_{n^*}$ based on Eq. (2), BetaCDF the cumulative distribution function (CDF) of Beta, and $\mathbb{1}$ the indicator function. Then, we have:

$$\delta_b = \text{BetaCDF}(p_b; \beta, \beta)^\gamma, \quad \beta, \gamma > 0, \quad (4)$$

$$p_b = \max_{\Delta t \in \{1, \ldots, T - t_{n^*}\}} \mathbb{E}_{y_{n, t_{n^*} : T} \sim f(\cdot | x_{n^*}, \mathcal{C})} \left[ \mathbb{1}\left( U\left(b + \Delta t, \tilde{y}_{b + \Delta t}\right) > U_{\text{prev}} \right) \right]. \quad (5)$$

$p_b$ in Eq. (5) is the probability that the current best configuration $x_{n^*}$ improves on $U_{\text{prev}}$ in some future BO step (i.e., probability of improvement, or PI (Mockus et al., 1978)). Intuitively, we want to defer the termination as $p_b$ increases, and vice versa. It is considered in Eq. (4) – as $p_b$ increases, the threshold $\delta_b$ increases as well because $\text{BetaCDF}(\cdot; \beta, \beta)^\gamma$ is a monotonically increasing function in $[0, 1]$, so we have less motivation to stop according to Eq. (3).

Fig. 3 plots $\text{BetaCDF}(\cdot; \beta, \beta)^\gamma$ in Eq. (4) over the various values of $\beta$. We can see that the function becomes vertical as $\beta \to +\infty$ and horizontal as $\beta \to 0$. In the former case, we terminate the BO

process when $p_b < 0.5$, ignoring the regret on the left-hand side of Eq.(3), whereas in the latter case, we ignore $p_b$ and decide solely based on the regret, with the threshold $\delta_b$ fixed to some value specified by $\gamma$ (e.g., in Fig. 3, $\delta_b = 0.2$ corresponds to $\gamma = \log_2 5$)[2]. Thus, the role of $\beta$ is to smoothly interpolate between the two extreme stopping criteria: 1) regret-based criterion (whether Eq. 3 is satisfied with $\delta_b = 0.2$ or not), and 2) PI-based criterion (whether $p_b > 0.5$ or not).

**Algorithm.** We summarize the pseudocode of our overall method in Alg. 1, with the red parts corresponding to the specifics of our method.

### 3.3 TRANSFER LEARNING OF LC EXTRAPOLATION

Since users may want to early-stop the BO, we should have a sample efficient LC extrapolation for preventing inaccurate early-stopping at the early stage of BO. We thus propose to use transfer learning to maximally improve the sample efficiency of our LC extrapolator.

**Transfer learning with LC mixup.** Among many plausible options, in this paper we propose to use Prior Fitted Networks (PFNs) (Müller et al., 2021) for LC extrapolation. PFNs are an in-context Bayesian inference method based on Transformer architectures (Vaswani et al., 2017), and show good performances on LC extrapolation (Adriaensen et al., 2023; Rakotoarison et al., 2024) without the computationally expensive online retraining (Kadra et al., 2023). A major difficulty of using PFNs for our purpose is that their training examples are generated only from a prior distribution, and to our knowledge, there are no existing ways to train PFNs with the given datasets. Also, PFNs require relatively a large Transformer architecture as well as huge amounts of training examples for good generalization performance (Adriaensen et al., 2023), which makes it risky to train PFNs with a finite set of examples.

Here we explain our novel transfer learning method for PFNs that can circumvent those difficulties with the mixup strategy (Zhang et al., 2018). Suppose we have $M$ different datasets and the corresponding $M$ sets of LCs collected from $N$ hyperparameter configurations. Define $l_{m,n} = (y_{n,1}^m, \ldots, y_{n,T}^m)$, the $T$-dimensional row vector of validation performances ($y$'s) collected from the $m$-th dataset and the $n$-th configuration, forming a complete LC of length $T$. Further define the matrix $L_m = [l_{m,1}^\top; \ldots; l_{m,N}^\top]^\top$, the row-wise stack of those LCs. In order to augment training examples, we propose to perform the following two consecutive mixups (Zhang et al., 2018):

1. Across datasets: $L' = \lambda_1 L_m + (1 - \lambda_1) L_{m'}$, with $\lambda_1 \sim \text{Unif}(0, 1)$, for all $m, m' \in [M]$.
2. Across configurations: $(x'', l'') = \lambda_2(x_n, l'_n) + (1 - \lambda_2)(x_{n'}, l'_{n'})$
   with $l'_n$ the $n$-th row of $L'$, $\lambda_2 \sim \text{Unif}(0, 1)$, for all $L'$ and $n, n' \in [N]$.

In this way, we can sample infinitely many training examples $\{(x'', l'')\}$ by interpolating between the LCs, leading to a robust LC extrapolator with less overfitting. Note that in the first step, we do not individually perform the mixup over the configurations but apply the same $\lambda_1$ to all the configurations, in order to preserve the correlations between the configurations encoded in the given datasets.

As for the network architecture and the training objective, we mostly follow Rakotoarison et al. (Rakotoarison et al., 2024). We use a similar Transformer architecture that takes a set of partial LCs and the corresponding configurations as an input and extrapolates the remaining part of the LCs. The training objective then maximizes the likelihood of those predictions conditioned on the partial LCs. We defer more details on the training to §E. Also, see §G for more discussion about the connection of our transfer learning method with ifBO (Rakotoarison et al., 2024) TNPs (Nguyen & Grover, 2022).

## 4 EXPERIMENTS

We next validate the efficacy of our method on various multi-fidelity HPO settings. We will publicly release our code upon acceptance.

**Datasets.** We use the following benchmark datasets for multi-fidelity HPO. **LCBench** (Zimmer et al., 2021): A LC dataset that evaluates the performance of 7 different hyperparameters on 35

---

[2]Note that the PI criterion in Eq. (5) is based on our novel acquisition function with utility. Therefore, the baselines should resort to only the regret-based criterion in Eq. (3). We found that $\delta_b = 0.2$ performs well over all the baselines, which corresponds to $\gamma = \log_2 5$ and $\beta \to 0$. Our method also use $\gamma = \log_2 5$ for fair comparison, but is allowed to use different $\beta > 0$ to combine it with the PI-based criterion in Eq. (5).

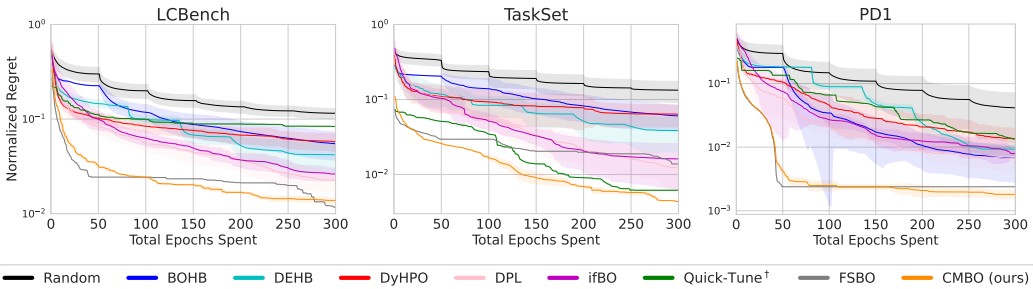

Figure 4: **The results on the conventional multi-fidelity HPO setup** ($\alpha = 0$). For each benchmark, we report the normalized regret of utility aggregated over all the test datasets.

different tabular datasets. The LCs are collected by training MLPs with 2,000 hyperparameter configurations, each for 51 epochs. We train our LC prediction model on 20 datasets and evaluate on the remaining 15 datasets. **TaskSet** (Metz et al., 2020): A LC dataset that consists of a diverse set of 1,000 optimization tasks drawn from various domains. We select 30 natural language processing (text classification and language modeling) tasks, train our LC extrapolator on 21 tasks, and evaluate on the remaining 9 tasks. Each task include 8 different hyperparameters and 1,000 their configurations. Each LC is collected by training models for 50 epochs. **PD1** (Wang et al., 2021): A LC benchmark that includes the performance of modern neural architectures (including Transformers) run on large vision datasets such as CIFAR-10, CIFAR-100 (Krizhevsky et al., 2009), ImageNet (Russakovsky et al., 2015), as well as statistical modeling corpora and protein sequence datasets from bioinformatics. We use 23 tasks with 4 different hyperparameters from SyneTune (Salinas et al., 2022) package, train our LC extrapolator on 16 tasks, and evaluate on the remaining 7 tasks. To facilitate transfer learning, we preprocess the data by excluding hyperparameter configurations with their training diverging (e.g., LCs with NaN), and linearly interpolate the LCs to match their length across different tasks. We then obtain the LCs of 50 epochs over the 240 configurations. See §D for more details.

**Baselines.** We compare our method against **Random Search** (Bergstra & Bengio, 2012) that randomly selects hyperparameter configurations sequentially. We next compare against several variants of Hyperband (Li et al., 2018) such as **BOHB** (Falkner et al., 2018) which replaces its random sampling of configurations with BO, and **DEHB** (Awad et al., 2021) which promotes internal knowledge transfer with evolution strategy. We also compare against more recent multi-fidelity BO methods such as **DyHPO** (Wistuba et al., 2022) which uses deep kernel GP (Wilson et al., 2016) and a greedy acquisition function with a short-horizon LC extrapolation, and **DPL** (Kadra et al., 2023) which extrapolates LCs with power law functions and model ensemble. **ifBO** (Rakotoarison et al., 2024) is an extension of PFNs (Müller et al., 2021) for freeze-thaw BO, whose acquisition is based on the PI at randomly chosen future training epochs. **Quick-Tune**[†], is a modified version of Quick-Tune (Arango et al., 2023) which is originally developed for dynamically selecting both pretrained models and hyperparmater configurations, with the additional cost term penalizing the non-uniform evaluation wall-time associated with each joint configuration. Since our experimental setup does not consider selecting pretrained models nor non-uniform evaluation wall-time, we only leave the transfer learning part of the model, which corresponds to a transfer learning version of DyHPO, i.e., we train its surrogate function with the same LC datasets used for training our LC extrapolator. Lastly, we compare against **FSBO** (Wistuba & Grabocka, 2020), a black-box transfer-BO that uses the same LC datasets to train a deep kernel GP surrogate. The difference of FSBO from Quick-Tune[†] is that its surrogate models the validation performances at the last epoch, whereas that of Quick-Tune[†] predicts the performances at the next epoch for multi-fidelity HPO. See §F for more details.

**Utility function.** While it is possible to collect user preference data manually and estimate the corresponding utility function (§3.1), in our experiments we use either linear function (i.e., $U(b, \tilde{y}) = \tilde{y} - \alpha b$), quadratic, square root, or stair-case function for simplicity. Note that for linear function, we let $\alpha \in \{0, 4e\text{-}05, 2e\text{-}04\}$, where $\alpha = 0$ does not penalize the number of BO steps at all – the BO does not terminate until the last BO step $B$ as with the conventional multi-fidelity BO setup.

**Stopping criterion.** As mentioned in the footnote in page 6, for the baselines we simply set the threshold $\delta_b = 0.2$ in Eq. (3). For our model, we also use $\gamma = \log_2 5$ for fair comparison, but use $\beta = e^{-1}$ for all the experiments in this paper, except the ablation study in Fig. 7d.

**Evaluation metric.** In order to report the average performances over the tasks, we use the normalized regret of utility $(U_{\max} - U_{b^*})/(U_{\max} - U_{\min}) \in [0, 1]$, similarly to Eq. (3). $U_{b^*}$ is the utility

Table 1: **Results on the cost-sensitive multi-fidelity HPO** with linear utility ($\alpha \in \{4e\text{-}05, 2e\text{-}04\}$). For better readability, we multiply 100 to the normalized regrets. Transfer learning methods are indicated by underline.

| Method | LCBench $\alpha = 4e\text{-}05$ Regret | Rank | LCBench $\alpha = 2e\text{-}04$ Regret | Rank | TaskSet $\alpha = 4e\text{-}05$ Regret | Rank | TaskSet $\alpha = 2e\text{-}04$ Regret | Rank | PD1 $\alpha = 4e\text{-}05$ Regret | Rank | PD1 $\alpha = 2e\text{-}04$ Regret | Rank |
|---|---|---|---|---|---|---|---|---|---|---|---|---|
| Random (Bergstra & Bengio, 2012) | $13.5_{\pm 2.3}$ | 8.1 | $17.9_{\pm 1.7}$ | 8.0 | $18.4_{\pm 4.8}$ | 7.7 | $22.3_{\pm 4.1}$ | 7.7 | $5.3_{\pm 2.9}$ | 7.1 | $9.6_{\pm 4.0}$ | 6.8 |
| BOHB (Falkner et al., 2018) | $7.0_{\pm 1.8}$ | 5.2 | $11.8_{\pm 1.7}$ | 5.5 | $8.0_{\pm 2.3}$ | 6.9 | $11.7_{\pm 2.1}$ | 6.5 | $1.8_{\pm 0.4}$ | 5.4 | $5.0_{\pm 0.3}$ | 5.1 |
| DEHB (Awad et al., 2021) | $5.7_{\pm 1.4}$ | 5.2 | $10.6_{\pm 1.2}$ | 5.6 | $5.3_{\pm 1.8}$ | 6.4 | $9.7_{\pm 1.4}$ | 6.2 | $2.1_{\pm 0.2}$ | 6.6 | $5.4_{\pm 0.2}$ | 6.7 |
| DyHPO (Wistuba et al., 2022) | $7.2_{\pm 1.2}$ | 6.1 | $12.1_{\pm 1.6}$ | 6.3 | $7.5_{\pm 2.1}$ | 7.0 | $11.1_{\pm 2.0}$ | 6.9 | $2.5_{\pm 0.6}$ | 6.3 | $6.2_{\pm 0.9}$ | 6.7 |
| DPL (Kadra et al., 2023) | $3.8_{\pm 0.5}$ | 3.6 | $9.3_{\pm 0.5}$ | 4.7 | $2.6_{\pm 0.7}$ | 4.0 | $7.5_{\pm 0.6}$ | 4.8 | $1.8_{\pm 0.3}$ | 4.4 | $5.1_{\pm 0.6}$ | 4.1 |
| ifBO (Rakotoarison et al., 2024) | $4.2_{\pm 0.4}$ | 3.8 | $9.3_{\pm 0.4}$ | 4.6 | $3.5_{\pm 1.2}$ | 4.4 | $8.1_{\pm 0.7}$ | 5.2 | $2.0_{\pm 0.1}$ | 5.7 | $5.8_{\pm 0.6}$ | 6.3 |
| Quick-Tune† (Arango et al., 2023) | $9.6_{\pm 0.0}$ | 6.9 | $12.7_{\pm 0.0}$ | 6.4 | $3.7_{\pm 0.0}$ | 3.9 | $5.6_{\pm 0.0}$ | 3.2 | $2.4_{\pm 0.0}$ | 5.4 | $5.5_{\pm 0.0}$ | 4.9 |
| FSBO (Wistuba & Grabocka, 2020) | $2.6_{\pm 0.0}$ | **2.9** | $6.4_{\pm 0.0}$ | 2.6 | $2.9_{\pm 0.0}$ | 3.2 | $4.9_{\pm 0.0}$ | 2.8 | $1.3_{\pm 0.0}$ | 2.2 | $4.2_{\pm 0.0}$ | 2.0 |
| **CMBO (ours)** | $\mathbf{2.3_{\pm 0.1}}$ | 3.2 | $\mathbf{3.1_{\pm 0.0}}$ | **1.3** | $\mathbf{1.3_{\pm 0.0}}$ | **1.6** | $\mathbf{3.1_{\pm 1.0}}$ | **1.7** | $\mathbf{0.8_{\pm 0.0}}$ | 2.1 | $\mathbf{0.9_{\pm 0.0}}$ | **1.0** |

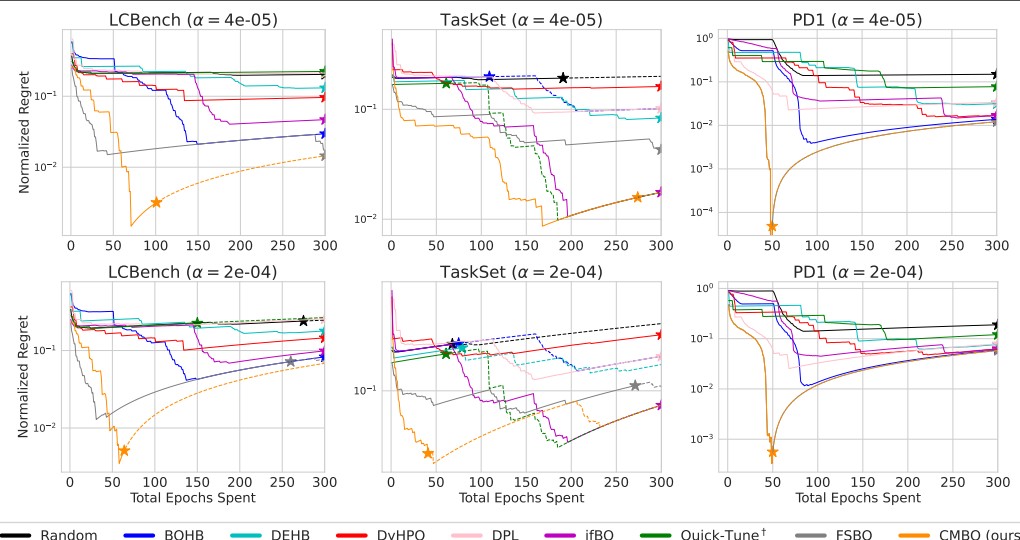

Figure 5: **Visualization of the normalized regrets over BO steps**. The first and second row correspond to $\alpha = 4e\text{-}05$ and $2e\text{-}04$, respectively. Each column corresponds to the cherry-picked examples from each benchmark. The asterisks indicate the stopping points, and the dotted lines represent the normalized regrets achievable by running each method without stopping. See §H for the results on all the other tasks.

obtained right after the BO terminates at step $b^*$, and $U_{\max}$ is the maximum achievable by running a single optimal configuration up to its maximum utility. Computing the exact $U_{\min}$ is a difficult combinatorial optimization problem, thus we simply approximate it with $U(B, y_1^{\mathrm{worst}})$, where $y_1^{\mathrm{worst}}$ is the worst 1-epoch validation performance across the configurations – we simply let $y_1^{\mathrm{worst}}$ decay over the maximum BO steps $B$, corresponding to a lower bound of the exact $U_{\min}$. We then average the normalized regrets across all the tasks in each benchmark, and report the mean and standard deviation over 5 runs, or even 30 runs for the baselines with relatively large variances such as Random, BOHB, DEHB. Lastly, we also report the rank of each method averaged over the tasks.

## 4.1 ANALYSIS

**Effectiveness of our transfer learning.** We first demonstrate the effectiveness of our transfer learning method. Fig. 4 shows the results on the conventional multi-fidelity HPO setting where we do not penalize the cost of BO at all ($\alpha = 0$). First of all, note that FSBO, a black-box transfer-BO method which switches its configuration only after a single complete training (e.g., 50 epochs), even outperforms all the other multi-fidelity methods that can change the configurations every epoch. The results clearly show the importance of transfer learning for improving the sample efficiency of HPO. Quick-Tune†, a

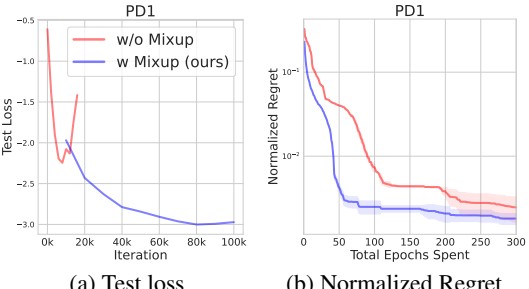

Figure 6: Ablation study on the mixup training. We use $\alpha = 0$ and PD1 benchmark for the experiments.

transfer version of DyHPO, performs similarly to the other baselines despite the use of the transfer learning, except on TaskSet. We attribute it to its greedy acquisition function, and more importantly

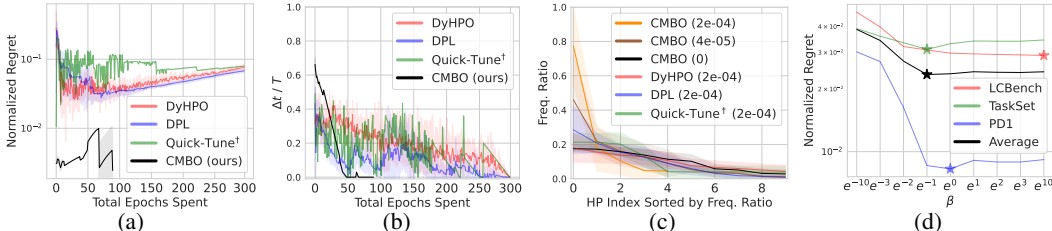

Figure 7: **(a, b, c)** Additional analysis on the effectiveness of our acquisition function. We use PD1 for the visualization. In **(c)**, the values of $\alpha$ are shown in the parenthesis. **(d)** Ablation study on $\beta$, with the minimum regret shown with the asterisks.

its lack of data augmentation. On the other hand, our method is non-greedy (when $\alpha = 0$) and can effectively augment the data with our mixup strategy, thereby showing significantly better performances than all the other multi-fidelity methods. Fig. 6 shows the ablation study on our mixup training. Fig. 6a shows that we can effectively reduce the risk of overfitting by adding the mixup strategy. As a result, the performance of BO improves significantly (Fig. 6b). Lastly, our method outperforms FSBO on TaskSet and PD1, while achieving comparative results on LCBench.

**Effectiveness of our acquisition function.** Next, Table 1 shows the performance of each method on the cost-sensitive multi-fidelity HPO setup ($\alpha > 0$). We see that our method largely outperforms all the methods on all the settings, including the multi-fidelity HPO and the transfer-BO methods, in terms of both normalized regret and average rank. Notice, our method achieves better average rank as the penalty becomes stronger ($\alpha = 2e\text{-}04$). Fig. 5 visualizes the normalized regret over the course of BO, where our method achieves significantly lower regret upon termination. Our method tends to achieve the minimum regret earlier than the baselines, demonstrating its sample efficiency in searching good hyperparameter configurations by explicitly considering the utility during the BO.

In order to clearly understand the source of improvements, we next analyze the configurations chosen by each method. Specifically, for each BO step $b$, we run the configuration currently selected at step $b$ up to its last epoch $T$, and compute its minimum ground-truth regret achievable at some future step $b + \Delta t$ (Fig. 7a), as well as the corresponding optimal increment $\Delta t$ (Fig. 7b). In Fig. 7a, our method shows much lower minimum regret than the baselines. It means that our acquisition function in Eq. (2) works as intended, trying to select at each BO step the best configuration which is expected to maximally improve the utility in future. Fig. 7b shows that the configurations chosen by our method initially correspond to greater $\Delta t$ (i.e., non-greedy), but gradually to the smaller $\Delta t$ (i.e., greedy). It is because as the BO proceeds, the performance improvements of BO saturate, so the cost of BO quickly dominates the trade-off, leading to smaller $\Delta t$ even close to 0. Fig. 7c shows the distribution of the top-10 most frequently selected configurations during the BO. As expected, our method tends to focus only on a few configurations during the BO to maximize the short-term performances, especially when the penalty is stronger with greater $\alpha$. On the other hand, the baselines tend to overly explore the configurations even when the penalty is the strongest ($\alpha = 2e\text{-}04$).

**Effectiveness of our stopping criterion.** We analyze the effectiveness of our stopping criterion in Eq. (3), (4), and (5). Fig. 7d shows the normalized regret over the different values of $\beta$, a mixing coefficient between the two extreme stopping criteria, as discussed in §3.2. $\beta \to 0$ corresponds to the criterion used by the baselines which is only based on the estimated normalized regret, whereas $\beta \to +\infty$ corresponds to the hard thresholding only based on the PI. We can see that the optimal criterion is achieved by smoothly mixing between the two ($\beta = e^{-1}$), demonstrating the superiority of our stopping criterion to the one used by the baselines ($\beta \to 0$).

**Experimental results on various utility functions.** In real-world scenarios, any functional forms of utility can be defined by various users. To investigate the effectiveness of CMBO on various utility functions, we perform additional experiments on PD1 benchmark using three additional utility functions as follows: **1) Staircase**: $U(b, \tilde{y}_b) = \tilde{y}_b - \sum_i \hat{\alpha}_i \mathbb{1}(b \in \mathcal{I}_i)$ where $\mathbb{1}(\cdot)$ is an indicator function and $\mathcal{I}_i$ is the $i$-th interval, **2) Quadratic**: $U(b, \tilde{y}_b) = \tilde{y}_b - \hat{\alpha}b^2$, and **3) Square Root**: $U(b, \tilde{y}_b) = \tilde{y}_b - \hat{\alpha}b^{1/2}$. Here, $\hat{\alpha}$ is the scaled $\alpha$ so that it has a similar range to the $\alpha$ used in the linear utility function (e.g., for the Quadratic utility function, we let $\hat{\alpha} := \alpha \frac{B}{B^2}$). We randomly sample $\alpha$'s from the uniform distribution $\alpha \sim \mathcal{U}(1e\text{-}05, 5e\text{-}03)$ to verify that the choice of $\alpha$ does not affect the overall results. Furthermore, We estimate a utility function using ifBO by assuming that the user

Table 2: **Results on the cost-sensitive multi-fidelity HPO setups with various utility functions** ($\alpha \sim \mathcal{U}(1e\text{-}05, 5e\text{-}03)$ with staircase, quadratic, and square root function). We estimate a utility function using ifBO by assuming that user wants better tradeoff than ifBO, denoted as "Estimated". See §C for more details.

| Method | Estimated | | Staircase | | | | Quadratic | | | | Square Root | | | |
| | - | | $\alpha = 0.00292$ | | $\alpha = 0.00436$ | | $\alpha = 0.00076$ | | $\alpha = 0.00459$ | | $\alpha = 0.00283$ | | $\alpha = 0.00378$ | |
| | Regret | Rank | Regret | Rank | Regret | Rank | Regret | Rank | Regret | Rank | Regret | Rank | Regret | Rank |
|---|---|---|---|---|---|---|---|---|---|---|---|---|---|---|
| Random | $25.5_{\pm7.7}$ | 6.9 | $25.2_{\pm1.7}$ | 7.5 | $26.2_{\pm1.6}$ | 7.5 | $16.6_{\pm1.5}$ | 7.6 | $20.3_{\pm0.9}$ | 7.3 | $26.8_{\pm1.3}$ | 7.1 | $26.3_{\pm1.2}$ | 7.0 |
| BOHB | $9.2_{\pm0.9}$ | 5.1 | $20.8_{\pm0.9}$ | 5.1 | $22.4_{\pm0.9}$ | 5.2 | $12.1_{\pm0.6}$ | 5.2 | $17.4_{\pm0.6}$ | 5.4 | $19.9_{\pm0.9}$ | 5.5 | $21.2_{\pm0.3}$ | 5.3 |
| DEHB | $14.8_{\pm6.3}$ | 6.4 | $23.6_{\pm0.3}$ | 7.1 | $23.0_{\pm0.9}$ | 7.1 | $14.0_{\pm0.1}$ | 7.5 | $18.7_{\pm1.2}$ | 6.9 | $19.9_{\pm0.9}$ | 6.0 | $19.7_{\pm0.5}$ | 5.6 |
| DyHPO | $10.9_{\pm1.6}$ | 5.9 | $22.0_{\pm1.2}$ | 6.5 | $23.0_{\pm0.9}$ | 6.4 | $14.3_{\pm2.3}$ | 6.7 | $18.7_{\pm1.2}$ | 6.9 | $20.1_{\pm2.2}$ | 5.7 | $20.3_{\pm2.0}$ | 5.6 |
| DPL | $15.5_{\pm7.0}$ | 5.9 | $19.2_{\pm0.9}$ | 4.4 | $20.8_{\pm0.9}$ | 4.3 | $11.8_{\pm1.9}$ | 4.8 | $16.5_{\pm0.8}$ | 4.7 | $17.6_{\pm1.2}$ | 4.9 | $18.6_{\pm1.1}$ | 4.8 |
| ifBO | $11.2_{\pm1.5}$ | 6.3 | $20.9_{\pm1.5}$ | 5.7 | $22.8_{\pm1.1}$ | 5.7 | $14.3_{\pm1.8}$ | 7.0 | $18.1_{\pm1.1}$ | 6.4 | $23.7_{\pm2.5}$ | 7.3 | $25.3_{\pm0.4}$ | 7.7 |
| Quick-Tune† | $19.5_{\pm0.0}$ | 4.8 | $21.3_{\pm0.0}$ | 6.9 | $24.4_{\pm0.0}$ | 7.5 | $11.6_{\pm0.0}$ | 4.5 | $17.3_{\pm0.0}$ | 4.5 | $17.2_{\pm0.0}$ | 3.4 | $17.6_{\pm0.0}$ | 3.4 |
| FSBO | $7.4_{\pm0.0}$ | 2.8 | $16.7_{\pm0.0}$ | 2.4 | $19.1_{\pm0.0}$ | 2.4 | $8.0_{\pm0.0}$ | 2.4 | $14.4_{\pm0.0}$ | 2.3 | $16.8_{\pm0.0}$ | 4.6 | $18.8_{\pm0.0}$ | 5.0 |
| **CMBO (ours)** | $\mathbf{2.1_{\pm0.0}}$ | **1.0** | $\mathbf{2.0_{\pm0.2}}$ | **1.0** | $\mathbf{0.9_{\pm0.0}}$ | **1.0** | $\mathbf{0.7_{\pm0.0}}$ | **1.0** | $\mathbf{0.7_{\pm0.0}}$ | **1.0** | $\mathbf{5.1_{\pm0.1}}$ | **1.0** | $\mathbf{4.6_{\pm0.0}}$ | **1.0** |

wants to achieve better trade-off than the one obtained by ifBO, denoted as "Estimated". See §C for moore details. Table 2 shows that our CMBO consistently outperforms all the baselines on various utility functions, showing that the superiority of our method is not affected by the types of utility functions.

**Ablation Studies.** To evaluate the effectiveness of each component, we conduct ablation studies on the proposed stopping criterion ($p_b$), acquisition function (Acq.), and transfer learning (T.), with mixup strategy on the PD1 benchmark. For the stopping criterion, we either use the smoothly-mixed criterion with $\beta = e^{-1}$ as in our full method ($p_b$ ✓), or use the regret-based criterion with $\beta \to 0$, the one used by the baselines ($p_b$ ✗). For the acquisition

Table 3: **Results of ablation study** using PD1 benchmark ($\alpha \in \{0, 4e\text{-}05, 2e\text{-}04\}$).

| $p_b$ | Acq. | T. | $\alpha = 0$ | $\alpha = 4e\text{-}05$ | $\alpha = 2e\text{-}04$ |
|---|---|---|---|---|---|
| ✗ | ✗ | ✗ | $0.8_{\pm0.1}$ | $2.0_{\pm0.1}$ | $5.8_{\pm0.0}$ |
| ✗ | ✗ | ✓ | $\mathbf{0.2_{\pm0.0}}$ | $1.4_{\pm0.0}$ | $5.7_{\pm0.3}$ |
| ✗ | ✓ | ✓ | $\mathbf{0.2_{\pm0.0}}$ | $1.2_{\pm0.0}$ | $4.4_{\pm0.0}$ |
| ✓ | ✓ | ✓ | $\mathbf{0.2_{\pm0.0}}$ | $\mathbf{0.8_{\pm0.0}}$ | $\mathbf{0.9_{\pm0.0}}$ |

function, we either use Eq.(2) (Acq. ✓) or the acquisition function of ifBO (Rakotoarison et al., 2024) (Acq. ✗). For transfer learning, we either use our surrogate trained with the proposed mixup strategy (T. ✓) or the surrogate of ifBO (Rakotoarison et al., 2024) (T. ✗). The results in Table 3 show that the performance improves sequentially as each component is added, with more pronounced improvements under strong penalties ($\alpha = 2e\text{-}4$). Notably, the stopping criterion does not affect the results in the conventional setting ($\alpha = 0$).

**Effectiveness on the real-world HPO.** Lastly, we investigate the effectiveness of our method on real-world object-detection dataset, along with estimating the utility function from the user preference data. From the 10 different datasets from RoboFlow100 (Ciaglia et al., 2022), we collect 500 LCs of validation performances by training three different network architectures, such as ResNet-50 (He et al., 2016), HR-Net (Wang et al., 2020), MobileNetv2 (Sandler et al., 2018), with 4 different hyperparameters (batch size, learning rate, momentum, and weight decay factor). Based on this setting, we

Table 4: **Results on the cost-sensitive multi-fidelity HPO** ($\alpha = 0, 4e - 05, 2e - 04$) **setups with object detection datasets.**

| Method | $\alpha = 0$ | | $\alpha = 4e\text{-}05$ | | $\alpha = 2e\text{-}04$ | |
| | Regret | Rank | Regret | Rank | Regret | Rank |
|---|---|---|---|---|---|---|
| Random | $5.0_{\pm1.3}$ | 6.5 | $7.1_{\pm2.6}$ | 6.4 | $13.1_{\pm2.6}$ | 6.5 |
| BOHB | $3.2_{\pm1.0}$ | 5.2 | $4.8_{\pm1.0}$ | 5.3 | $10.7_{\pm1.0}$ | 5.4 |
| DEHB | $5.0_{\pm1.4}$ | 6.6 | $6.6_{\pm1.4}$ | 6.5 | $12.4_{\pm1.3}$ | 6.6 |
| DyHPO | $16.0_{\pm2.5}$ | 5.9 | $17.5_{\pm2.5}$ | 6.0 | $23.1_{\pm2.7}$ | 6.2 |
| DPL | $3.9_{\pm1.4}$ | 4.6 | $5.5_{\pm1.4}$ | 4.8 | $11.4_{\pm1.3}$ | 5.2 |
| ifBO | $2.3_{\pm0.5}$ | 4.3 | $3.9_{\pm0.5}$ | 4.3 | $9.8_{\pm0.5}$ | 4.4 |
| Quick-Tune† | $5.3_{\pm0.0}$ | 4.8 | $6.9_{\pm0.0}$ | 4.9 | $12.6_{\pm0.0}$ | 5.0 |
| FSBO | $2.1_{\pm0.0}$ | 3.9 | $3.7_{\pm0.0}$ | 3.9 | $9.6_{\pm0.0}$ | 4.2 |
| **CMBO (ours)** | $\mathbf{1.3_{\pm0.1}}$ | **3.3** | $\mathbf{3.6_{\pm0.3}}$ | **2.9** | $\mathbf{5.7_{\pm0.3}}$ | **1.4** |

construct 30 tasks (= 3 network architectures × 10 datasets) and split them into 20 / 10 tasks for meta-training / meta-test, respectively. In Table 4, we can clearly see that our method consistently and significantly outperforms all the baselines on this real-world dataset as well, on both the estimated utility function and the linear utility functions with various $\alpha$.

## 5 CONCLUSION

In this paper, we discussed cost-sensitive multi-fidelity BO, a novel framework for dramatically improving the efficiency of HPO. Based on the assumption that users want to early-stop the BO when the utility saturates, we explained how to achieve the maximum utility with our novel acquisition function and the stopping criterion specifically tailored to this problem setup, as well as the novel transfer learning method for training a sample efficient in-context LC extraploator. We empirically demonstrated the effectiveness of our method over the previous multi-fidelity HPO and the transfer-BO methods, with numerous empirical evidence strongly supporting our claim.

**Reproducibility statement.** All the implementation details are described throughout §D, E, and F. We provide anonymized code in supplemental materials, and will publish the code upon acceptance.

**Ethics statement.** Our work presents a cost-sensitive multi-fidelity Bayesian Optimization method designed to make HPO more accessible to users with limited computational resources. By allowing users to define a utility function that balances performance improvements with computational cost, our method enables more efficient optimization and early stopping when costs outweigh benefits. This approach helps under-resourced individuals or organizations achieve competitive results without excessive financial or computational burden, promoting inclusivity and reducing inequality in machine learning research and practice.

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

# A RELATED WORK

**Multi-fidelity HPO.** Unlike traditional black-box approaches for HPO (Bergstra & Bengio, 2012; Hutter et al., 2011; Bergstra et al., 2011; Snoek et al., 2012; 2015; 2014; Cowen-Rivers et al., 2022; Müller et al., 2023), multi-fidelity (or gray-box) HPO aims to optimize hyperparameters in a sample efficient manner by utilizing low fidelity information (e.g., validation set performances with smaller training dataset) as a proxy for higher or full fidelities (Swersky et al., 2013; Kandasamy et al., 2016; Klein et al., 2017a; Poloczek et al., 2017; Kandasamy et al., 2017; Wu & Frazier, 2018; Wu et al., 2020), dramatically speeding up the HPO. In this paper, we focus on making use of performances at fewer training epochs to better predict/optimize the performances at longer training epochs. One of the well-known examples is Hyperband (Li et al., 2018), a bandit-based method that randomly selects a set of random hyperparameter configurations, and stops poorly performing ones using successive halving (Karnin et al., 2013) even before reaching the last training epoch. While Hyperband shows much better performance than random search (Bergstra & Bengio, 2012), its computational or sample efficiency can be further improved by replacing random sampling of configurations with Bayesian optimization (Falkner et al., 2018), adopting evolution strategy to promote internal knowledge transfer (Awad et al., 2021), or making it asynchronously parallel (Li et al., 2020).

**Transfer BO.** Transfer learning can be used for improving the sample efficiency of BO (Bai et al., 2023), and here we list a few of them. Some of recent works explore scalable transfer learning with deep neural networks (Perrone et al., 2018; Wistuba & Grabocka, 2020). Also, different components of BO can be transferred such as observations (Swersky et al., 2013), surrogate functions (Golovin et al., 2017; Wistuba & Grabocka, 2020), hyperparmater initializations (Wistuba & Grabocka, 2020), or all of them (Wei et al., 2021). However, most of the existing transfer-BO approaches assume the traditional black-box BO settings. To the best of our knowledge, Quick-Tune (Arango et al., 2023) is the only recent work which targets multi-fidelity and transfer BO at the same time. However, as mentioned above, their multi-fidelity BO formulation is greedy, whereas our transfer-BO method can dynamically control the degree of greediness during the BO by explicitly taking into consideration the trade-off between cost and performance of BO.

**Cost-sensitive HPO.** Multi-fidelity BO is inherently cost-sensitive since predictions get more accurate as the gap between the fidelities becomes closer. However, the performance metric of such vanilla multi-fidelity BO monotonically increases as we spend more budget, whereas in this paper we want to find the optimal trade-off between the amount of budget spent thus far and the corresponding intermediate performances of BO, thereby automatically early-stopping the BO around the maximal utility. Quick-Tune (Arango et al., 2023) also suggests a cost-sensitive BO in multi-fidelity settings, but unlike our work, their primary focus is to trade-off between the performance and the cost of BO associated with pretrained models of various size, which can be seen as a generalization of more traditional notion of cost-sensitive BO (Snoek et al., 2012; Abdolshah et al., 2019; Lee et al., 2020), from black-box to multi-fidelity settings. In addition to the above discussion, Makarova et al. (2022) propose a stopping criterion which terminates the BO when the suboptimality in optimizing validation performance (instead of the test performance) is dominated by the statistical estimation error. Roughly speaking, this stopping criterion can be seen as an instance of our utility-based stopping criteria, where the user preference is not willing to spend further computational budgets after reaching the certain BO performance. Furthermore, in this paper we focus on maximizing any possible utility functions defined by user, instead of minimizing population error.

**BO with user preference.** Several works have tried to encode user's initial belief on good hyperparameter configurations into BO frameworks (Souza et al., 2021; Hvarfner et al., 2021; Mallik et al., 2024). On the othet hand, our paper suggests encoding user's preference about the trade-off between cost and performance of multi-fidelity BO. Therefore, the notion of user preference in this paper is largely different from the previous literature.

# B    UTILITY ESTIMATION

As mentioned in the main paper, it is not easy for users to define or quantify their utilities. Here we first briefly remind the notion of utility and then detail how we simulate the estimation of user utility with Bradley-Terry (BT) model  (Bradley & Terry, 1952).

**Utility.**    A utility function $U$ describes the trade-off between the BO step $b$ and the BO performance $\tilde{y}_b$. Its values $U(b, \tilde{y}_b)$ negatively correlate with $b$ and positively with $\tilde{y}_b$. For instance, we can assume a linear utility function $U(b, \tilde{y}_b) = \tilde{y}_b - \alpha b$ for some $\alpha > 0$, such that the utility gives linear incentives and penalties to the performance and number of BO steps, respectively.

**Functional forms.**    In real-world scenario, however, one can have much more complex utility function other than the above linear case. We therefore consider the following additional functional forms including linear one: **1) Linear**: $U^{(\text{linear})}(b, y) = \tilde{y} - \alpha b$, **2) Staircase**: $U^{(\text{stair})}(b, \tilde{y}_b) = \tilde{y}_b - \sum_i \alpha_i \mathbb{1}(b \in A_i)$ ($\mathbb{1}(\cdot)$ is an indicator function, and $A_i$ is an $i$-th interval), **3) Quadratic**: $U^{(\text{quad})}(b, \tilde{y}_b) = \tilde{y}_b - \alpha b^2$, and **4) Square Root**: $U^{(\text{sqrt})}(b, \tilde{y}_b) = \tilde{y}_b - \alpha b^{1/2}$. In contrast to the notation of main paper, we normalize the budget $b$ by the allowed total computation budget $B$, i.e., $b \in [0, 1]$ instead of $b \in \{1, \dots, B\}$. Furthermore, we assume that these utility functions can be linearly combined, e.g., $U = w^{(\text{linear})} U^{(\text{linear})} + \dots + w^{(\text{sqrt})} U^{(\text{sqrt})}$, where $w^{(\text{linear})} + \dots + w^{(\text{sqrt})} = 1$.

**Data collection.**    We now describe how we roughly estimate user utility based on the user preference data pairs. First of all, we assume that it is possible for users to decide whether they prefer one point to the other one, instead of quantifying their utility, i.e., we can collect user preference data. For simulation, we assume that we are given these preference data generated by true utility function. True utility function $U$ is randomly defined by sampling penalizing coefficient from $\mathbb{U}(0, 1)$ and linear combination coefficients from Dirichlet distribution. We randomly select *meaningful* data pairs from $b \sim \mathbb{U}(0, 1)$ and $\tilde{y}_b \sim \mathbb{U}(0, 1)$. Here, meaningful data pairs means that one datapoint of each pair is not trivially preferred by user, for example, one has larger performance $\tilde{y}_b$ with smaller budget $b$ than the other. A user then label their preference on these data pairs; for simulation, we label them by using the true utility functions.

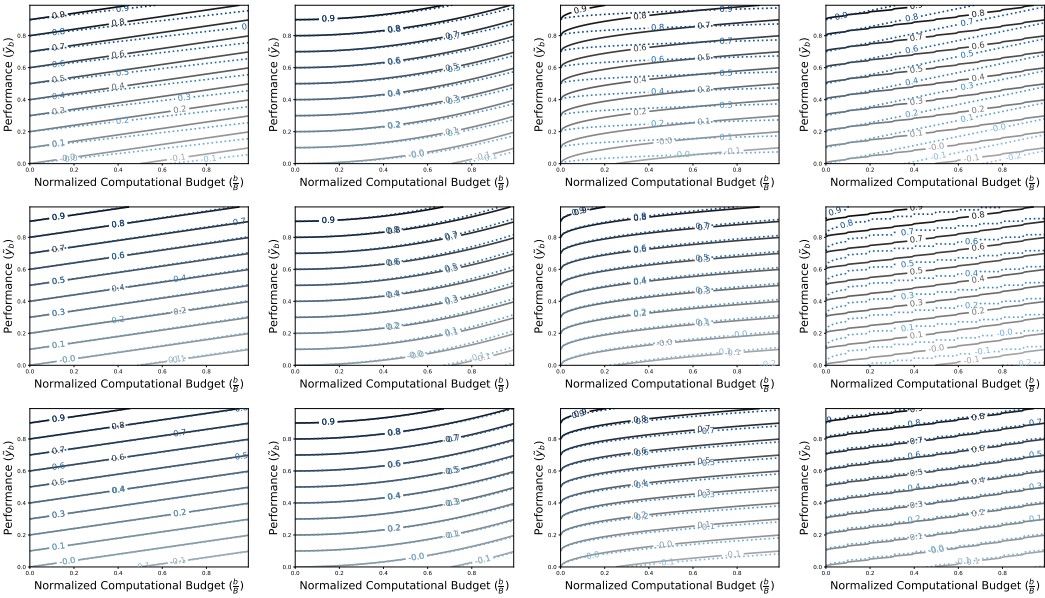

Figure 8: **Contour plots of true utilities and their approximations**. From left to right, the columns show **different functional forms** of linear, quadratic, square root, and a combination of four different functions including a staircase function. From top to bottom, the rows represent **30, 100, and 1000 user preference data pairs**.

**Training details and results.** As explained in the main paper, we use binary cross entropy loss between the probability of preference described by the BT model in Eq. (1) and the label. In Fig. 2 of the main paper, we set $w^{(staircase)}$ to be 0. We begin by randomly initializing another utility function to approximate a randomly sampled true utility function, setting the linear combination and penalizing coefficients to $\frac{1}{3}$ and 0.0001, respectively. We use gradient-based optimization algorithm (e.g., SGD, L-FBGS) with 1000 iterations for optimizing the coefficients. The temperature term $\tau$ in Eq. (1) is set to 0.05.

In addition to Fig. 2, we perform other experiments on single utility functions such as linear, quadratic, square root, and combination of the four functional forms we consider. Fig. 8 demonstrates that not only can single utilities –linear, quadratic, and square root – be well approximated using preference data, but even more complex utilities (e.g., a combination of four different utilities) can also be accurately approximated. Furthermore, we found that the approximation works well even with smaller numbers (e.g., 30, 100) of user preference data pairs for simple cases.

## C UTILITY FUNCTION ESTIMATION FOR PD1

In this section, we explain the experimental details for utility estimation denoted as "Estimated" in Table 2. Here, we assume that the user wants to set the trade-off (between the cost and performance of BO) to the trade-off achievable by running other multi-fidelity HPO methods, such as ifBO (Rakotoarison et al., 2024). Therefore, we run ifBO to all the meta-training tasks and average those BO trajectories, obtaining a single BO trajectory corresponding to the overall representative trade-off on the PD1 dataset.

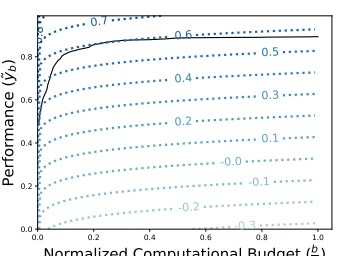

Figure 9: **The estimated utility function with the user preference data.** The solid line represents the average BO trajectory obtained by running ifBO on the meta-training tasks, and the dotted lines show the correspondingly estimated utility function.

Based on that single curve, we randomly sample many points around that curve, such that all the points locate either upper or bottom parts of the curve. Then, we can collect infinitely many pair of points by randomly picking one upper point and one bottom point (with the constraint that the upper one should be on the right side of the bottom one), and construct the user preference data $\{(u_1, u_2)\}$. As a fitting function, we use an exponential function with bias term, i.e., $\tilde{U}(b, \tilde{y}_b) = \tilde{y}_b - \alpha b^a + c$, where $\alpha, a, c > 0$.

Figure 9 shows the average BO trajectory obtained by running BOHB on the meta-training tasks (solid black line), and the correspondingly estimated utility function (dotted blue lines). We can see that the shape of the utility function fits reasonably well to the trajectory of ifBO.

## D DETAILS ON BENCHMARKS AND DATA PREPROCESSING

In this section, we elaborate the details on the LC benchmarks and data preprocessing we have done.

**LCBench** We use [APSFailure, Amazon_employee_access, Australian, Fashion-MNIST, KDD-Cup09_appetency, MiniBooNE, adult, airlines, albert, bank-marketing, blood-transfusion-service-center, car, christine, cnae-9, connect-4, covertype, credit-g, dionis, fabert, helena] for training LC extrapolator. We evaluate it on [higgs, jannis, jasmine, jungle_chess_2pcs_raw_endgame_complete, kc1, kr-vs-kp, mfeat-factors, nomao, numerai28.6, phoneme, segment, shuttle, sylvine, vehicle, volkert]. Each task contains 2000 LCs with 51 training epochs. We summarize the hyperparameter of LCBench in Table 5.

Table 5: The 7 hyperparameters for **LCBench** tasks.

| Name | Type | Vaules | Info |
|---|---|---|---|
| batch_size | integer | $[16, 51]$ | log |
| learning_rate | continuous | $[0.0001, 0.1]$ | log |
| max_dropout | continuous | $[0.0, 1.0]$ | |
| max_units | integer | $[64, 1024]$ | log |
| momentum | continuous | $[0.1, 0.99]$ | |
| max_layers | integer | $[1, 5]$ | |
| weight_decay | continuous | $[1e - 05, 0.1]$ | |

**TaskSet** We use [rnn_text_classification_family_seed{19, 3, 46, 47, 59, 6, 66}, word_rnn_language_model_family_seed{22, 47, 48, 74, 76, 81}, char_rnn_language_model_family_{seed19, 26, 31, 42, 48, 5, 74}] for training LC extrapolator. We evaluate it on [rnn_text_classification_family_seed{8, 82, 89}, word_rnn_language_model_family_seed{84, 98, 99}, char_rnn_language_model_family_seed{84, 94, 96}]. Each task contains 1000 LCs with 50 training epochs. We summarize the hyperparameter of TaskSet in Table 6.

Table 6: The 8 hyperparameters for **Taskset** tasks.

| Name | Type | Vaules | Info |
|---|---|---|---|
| learning_rate | continuous | $[1e - 09, 10.0]$ | log |
| beta1 | continuous | $[0.0001, 1.0]$ | |
| beta2 | continuous | $[0.001, 1.0]$ | |
| epsilon | continuous | $[1e - 12, 1000]$ | log |
| l1 | continuous | $[1e - 09, 10.0]$ | log |
| l2 | continuous | $[1e - 09, 10.0]$ | log |
| linear_decay | continuous | $[1e - 08, 0.0001]$ | log |

**PD1** We use [ uniref50_transformer_batch_size_128, lm1b_transformer_batch_size_2048, imagenet_resnet_batch_size_256, mnist_max_pooling_cnn_tanh_batch_size_2048, mnist_max_pooling_cnn_relu_batch_size_{2048, 256}, mnist_simple_cnn_batch_size_{2048, 256}, fashion_mnist_max_pooling_cnn_tanh_batch_size_2048, fashion_mnist_max_pooling_cnn_relu_batch_size_{2048, 256}, fashion_mnist_simple_cnn_batch_size_{2048, 256}, svhn_no_extra_wide_resnet_batch_size_1024, cifar{100, 10}_wide_resnet_batch_size_2048] for training LC extrapolator. We evaluate it on [imagenet_resnet_batch_size_512, translate_wmt_xformer_translate_batch_size_64, mnist_max_pooling_cnn_tanh_batch_size_256, fashion_mnist_max_pooling_cnn_tanh_batch_size_256, svhn_no_extra_wide_resnet_batch_size_256, cifar100_wide_resnet_batch_size_256, cifar10_wide_resnet_batch_size_256]. Each task contains 240 LCs with 50 training epochs. We summarize the hyperparameter of PD1 in Table 7.

Table 7: The 8 hyperparameters for **PD1** tasks.

| Name | Type | Vaules | Info |
|---|---|---|---|
| lr_initial_value | continuous | $[1e - 05, 10.0]$ | log |
| lr_power | continuous | $[0.1, 2.0]$ | |
| lr_decay_steps_factor | continuous | $[0.01, 0.99]$ | |
| one_minus_momentum | continuous | $[1e - 05, 1.0]$ | log |

**Data Preprocessing** As will be detailed in the §F, we use the 0-epoch LC value $y_{n,0}$ which is the performance before taking any gradient steps. The 0-epoch LC values originally are not provided except for LCBench; we use the log-loss of the first epoch as the 0-epoch LC value for TaskSet, as it is already sufficiently large in our chosen tasks. For PD1, we interpolate the LCs to be the length of 51 training epochs, and we take the first performance as the 0-epoch LC value. Furthermore, we take the average over the 0-epoch LC values $\bar{y}_0$ since it is hard to have different initial values among optimizer hyperparameter configurations in a task, without taking any gradient steps. For

transfer learning, we follow the convention of PFN (Adriaensen et al., 2023) for data preprocessing; we consistently apply non-linear LC normalization[3] to the LC data of three benchmarks, which not only maps either accuracy or log-loss LCs into $[0, 1]$ but also simply make our optimization as a maiximization problem. To facilitate transfer learning, we use the maximum and minimum values in each task in LCBench and PD1 benchmark for the LC normalization. In TaskSet, we only use the $\bar{y}_0$ for the LC normalization.

## E DETAILS ON ARCHITECTURE AND TRAINING OF LC EXTRAPOLATOR

In the section, we elaborate our LC extrapolator model and how to train it on the learning curve dataset.

**Construction of Context and Query points.** As mentioned earlier in §3.3, the whole training pipeline of our learning curve extrapolator model can be seen an instance of TNPs (Nguyen & Grover, 2022). Here we can simulate each step of Bayesian Optimization; predicting the remaining part of LC in all configurations conditioned on the set $\mathcal{C}$ of the collected partial LCs. To do so, we construct a training task by randomly sampling context and query points from LC benchmark after the proposed LC mixup as follows:

1. We choose a LC dataset $L_m = [l_{m,1}^\top; \ldots; l_{m,N}^\top]^\top \in \mathbb{R}^{N \times T}$ by randomly sampling $m \in [M]$.

2. From $L_m$, we randomly sample $n_1, \ldots, n_C \in [N]$ and $t_1, \ldots, t_C \in [T]$ and construct context points of $X^{(c)} = [x_{n_1}^\top, \ldots, x_{n_C}^\top]^\top \in \mathbb{R}^{C \times d_x}$, $T^{(c)} = [t_1/T, \ldots, t_C/T]^\top \in \mathbb{R}^{C \times 1}$, and $Y^{(c)} = [y_{n_1,t_1}, \ldots, y_{n_C,t_C}] \in \mathbb{R}^{C \times 1}$.

3. From $L_m$, we exclude $n_1, \ldots, n_C \in [N]$ and $t_1, \ldots, t_C \in [T]$ and randomly sample $n'_1, \ldots, n'_Q \in [N]$ and $t'_1, \ldots, t'_Q \in [T]$ and construct query points of $X^{(q)} = [x_{n'_1}^\top, \ldots, x_{n'_Q}^\top]^\top \in \mathbb{R}^{Q \times d_x}$, $T^{(q)} = [t'_1/T, \ldots, t'_C/T]^\top \in \mathbb{R}^{Q \times 1}$, and $Y^{(q)} = [y_{n'_1,t'_1}, \ldots, y_{n'_Q,t'_Q}] \in \mathbb{R}^{Q \times 1}$.

**Transformer for Predicting Learning Curves.** From now on, we denote each row vector of the constructed context and query points with the lowercase, e.g., $y^{(q)}$ of $Y^{(q)}$. We learn a Transformer-based learning curve extrapolator model which is a probabilistic model of $f(Y^{(q)}|X^{(c)}, T^{(c)}, Y^{(c)}, X^{(q)}, T^{(q)})$. Conditioned on any subsets of LCs (i.e., $X^{(c)}, T^{(c)}$, and $Y^{(c)}$), this model predicts a mini-batch of the remaining part of LCs of existing hyperparameter configurations in a given dataset (i.e., $Y^{(q)}$ of $X^{(q)}$ and $T^{(q)}$). For the computational efficiency, we further assume that the query points are independent to each other, as done in PFN (Adriaensen et al., 2023):

$$f(Y^{(q)}|X^{(c)}, T^{(c)}, Y^{(c)}, X^{(q)}, T^{(q)}) = \prod_{x^{(q)}, t^{(q)}, y^{(q)}} f(y^{(q)}|x^{(q)}, t^{(q)}, X^{(c)}, T^{(c)}, Y^{(c)}). \quad (6)$$

Before encoding the input into the Transformer, we first encode the input of $X^{(c)}, T^{(c)}, Y^{(c)}, X^{(q)}$, and $T^{(q)}$ using simple linear layer as follows:

$$H^{(c)} = X^{(c)} W_x + T^{(c)} W_t + Y^{(c)} W_y \quad (7)$$

$$H^{(q)} = X^{(q)} W_x + T^{(q)} W_t, \quad (8)$$

where $W_x \in \mathbb{R}^{d_x \times d_h}$, $W_t \in \mathbb{R}^{1 \times d_h}$, and $W_y \in \mathbb{R}^{1 \times d_h}$. Here, we abbreviate the bias term.

Then we concatenate the encoded represnetations of $H^{(c)}$ and $H^{(q)}$, and feedforward it into Transformer layer by treating each pair of each row vector of $H^{(c)}$ and $H^{(q)}$ as a separate position/token as follows:

$$H = \texttt{Transformer}([H^{(c)}; H^{(q)}, \texttt{Mask}]) \in \mathbb{R}^{(M+N) \times d_h} \quad (9)$$

$$\hat{Y} = \texttt{Head}(H) \in \mathbb{R}^{(M+N) \times d_o}, \quad (10)$$

---

[3]The details can be found in Appendix A of PFN (Adriaensen et al., 2023) and https://github.com/automl/lcpfn/blob/main/lcpfn/utils.py.

where `Transformer`$(\cdot)$ and `Head`$(\cdot)$ denote the Transformer layer and multi-layer perceptron (MLP) for the output prediction, respectively. `Mask` $\in \mathbb{R}^{(N_c+N_q)\times(N_c+N_q)}$ is the mask of transformer that allows all the tokens to attend context tokens only. Here, the output dimension $d_o$ is specified by output distribution of $y$. Following PFN (Adriaensen et al., 2023), we discretize the domain of $y$ by $d_o = 1000$ and use the categorical distribution. Finally, we only take the output of the last $N_q$ tokens as output, i.e., $\hat{Y}^{(q)} = \hat{Y}[:, N_c : (N_c + N_q)] \in \mathbb{R}^{N_q \times d_h}$ (PyTorch-style indexing operation), since we only need the outputs of query tokens for modeling $\prod f(y^{(q)}|x^{(q)}, t^{(q)}, X^{(c)}, T^{(c)}, Y^{(c)})$.

**Training Objective.** Our pre-training objective is then defined as follows:

$$\arg\min_f \mathbb{E}_p \left[ - \sum_{x^{(q)}, t^{(q)}, y^{(q)}} \log f(y^{(q)}|x^{(q)}, t^{(q)}, X^{(c)}, T^{(c)}, Y^{(c)}) \right] + \lambda_{\text{PFN}} \mathcal{L}_{\text{PFN}}, \qquad (11)$$

where $\mathcal{D}_{KL}$ is the Kullback–Leibler divergence, and $p$ is the empirical LC data distribution. We additionally minimize $\mathcal{L}_{\text{PFN}}$ with coefficient $\lambda_{\text{PFN}}$, which is the LC extrapolation loss in each LC (Adriaensen et al., 2023). We found $\lambda_{\text{PFN}} = 0.1$ works well for most cases. We use the stochastic gradient descent algorithm to solve the above optimization problem.

**Training Details.** We sample 4 training tasks for each iteration, i.e., the size of meta mini-batch is set to 4. We uniformly sample the size $C$ of context points from 1 to 300, and the size of query points $Q$ is set to 2048. Following PFN (Adriaensen et al., 2023), the hidden size of each Transformer block $d_h$, the hidden size of feed-forward networks, the number of layers of Transformer, dropout rate are set 1024, 2048, 12, 0.2. We use GeLU (Hendrycks & Gimpel, 2016). We train the extrapolator for 10,000 iterations on training split of each benchmark with Adam (Kingma & Ba, 2014) optimizer. The $\ell_2$ norm of meta mini-batch gradient is clipped to 1.0. The learning rate is linearly increased to 2e-05 for 25000 iterations, and it is decreased with a cosine scheduling until the end. The whole training process takes roughly 10 hours in one NVIDIA Tensor Core A100 GPU.

# F ADDITIONAL DETAILS ON EXPERIMENTAL SETUPS

In this section, we elaborate additional details on the experimental setups.

**0-epoch LC value.** We assume the access of the 0-epoch LC value $\bar{y}_0$ in §D which is the model performance before taking gradient steps. This is also plausible for realistic scenarios since in most deep-learning models one evaluation cost is acceptable in comparison to training costs. The 0-epoch LC value $\bar{y}_0$ is always conditioned on our LC extrapolator $f$ for both pretraining and BO stage.

**Monte-Carlo (MC) sampling for reducing variance of LCs.** As mentioned in §3.2, we estimate the expectation of proposed acquisition function $A$ in Eq. (2) with 1000 MC samples. We found that each LC $y_{n,t_{n:T}}$ sampled from LC extrapolator $f(\cdot|x_n, \mathcal{C})$ is noisy, due to the assumption that query points of $y_{n,t_{n:T}}$ are independent to each other in Eq. (6). We compute $\tilde{y}_{b+\Delta t}$ by taking the maximum among the last step BO performance (i.e., cumulative max operation), therefore, the quality of estimation highly degenerates due to the noise in the small $\Delta t$. To prevent this, we reduce the variance of MC samples by taking the average of the sampled LCs. For example, we sample 5000 LC samples from the LC extrapolator $f$, then we divide them into 1000 groups and take the average among the 5 LC samples in each group. We empirically found that this stabilize the estimation of not only acquisition function $A$ and probability of utility improvement $p_b$ in Eq. (5).

**Inference Time for BO.** In Table 8, we report average wall-clock time and standard deviation spent on BO over 5 runs for PDL, ifBO, CMBO. For ifBO, we use the same surrogate model provided in github but re-implement the BO process of ifBO based on our code base, which dramatically reduce the wall-clock time of original implementation. We measure all the wall-clock times in one NVIDIA Tensor Core A100 GPU using the same experimental setups. ifBO is the most of efficient method among

Table 8: **Wall-clock time (seconds) for BP on LCBench, TaskSet, and PD1 datasets**

| Method | LCBench | TaskSet | PD1 |
|---|---|---|---|
| DPL | $194.4_{\pm 6.15}$ | $191.5_{\pm 2.37}$ | $189.3_{\pm 1.73}$ |
| ifBO | $174.4_{\pm 2.80}$ | $90.4_{\pm 1.02}$ | $23.2_{\pm 1.16}$ |
| **CMBO (ours)** | $456.0_{\pm 5.06}$ | $234.3_{\pm 3.21}$ | $67.6_{\pm 1.85}$ |

them, but we believe that the difference between the wall-clock time of ifBO and CMBO is negligible since training neural networks usually dominates the total wall-clock time spent for HPO.

**Details on Baseline Implementation.** We list the implementation detils for baselines as follows:

1. **Random Search.** Instead of randomly selecting a hyperparameter configuration for each BO step, we run the selected configuration until the last epoch $T$.

2. **BOHB and DEHB.** We follow the most recent implementation of these algorithms in Quick-Tune (Arango et al., 2023). We slightly modify the official code[4], which is heavily based on SyneTune (Salinas et al., 2022) package.

3. **DPL.** We follow the official code[5] provided the authors of DPL (Kadra et al., 2023), and slightly modify the benchmark implementation to incorporate our experimental setups.

4. **ifBO.** We follow the official code[6] provided the authors for surrogate model of ifBO (Rakotoarison et al., 2024), and incorporate the surrogate model in our code base to be aligned with our experimental setups.

5. **DyHPO and Quick-Tune†.** We follow the official code[7] provided the authors of DyHPO (Wistuba et al., 2022), and slightly modify the benchmark implementation to incorporate our experimental setups. For Quick-Tune†, we pretrain the deep kernel GP for 50000 iterations with Adam optimizer with mini-batch size of 512. The initial learning rate is set to 1e-03 and decayed with cosine scheduling. To leverage the transfer learning scenario, we use the best configuration among the LC datasets which is used for training the GP as an initial guess of BO.

6. **FSBO.** FSBO does not provide an official code, therefore, we follow an available code in the internet[8]. We also slightly modify the benchmark implementation, and use the best configuration among the LC datasets as an initial guess.

## G CONNECTION BETWEEN OUR MIXUP STRATEGY WITH IFBO AND TNP

Our mixup strategy is reminiscent of the data generation scheme of ifBO (Rakotoarison et al., 2024), a variant of PFNs for in-context freeze-thaw BO. Similarly to our ancestral sampling, ifBO first samples random weights for a neural network (i.e., a prior distribution) to sample a correlation between configurations (the first mixup step), and then linearly combines a set of basis functions to generate LCs (the second mixup step). Our training method differs from ifBO in that our prior distribution is implicitly defined by LC datasets and the mixup strategy, whereas ifBO resorts to a manually defined distribution.

Indeed, our training method is more similar to Transformer Neural Processes (TNPs) (Nguyen & Grover, 2022), a Transformer variant of Neural Processes (NPs) (Garnelo et al., 2018). Similarly to PFNs, TNPs directly maximize the likelihood of target data given context data with a Transformer architecture, which differs from the typical NP variants that summarize the context into a latent variable and perform variational inference on it. Moreover, as with the other NP variants, TNPs meta-learn a model over a distribution of tasks to perform sample efficient probabilistic inference. In this vein, the whole training pipeline of our LC extrapolator can be seen as an instance of TNPs – we also meta-learn a sample efficient Transformer-based LC extrapolator over the distribution of LCs induced by the mixup strategy.

## H ADDITIONAL EXPERIMENTAL RESULTS

**Ablation Study on Cost Limits** To understand the behavior of our method under varying cost limits (or total computational budgets, i.e., $B$), we conducted additional experiments similar to those in Fig. 7b, varying the cost limits to 100, 200, and 300. Fig. 10 presents the distribution of the top-10 most frequently chosen hyperparameter configurations throughout the optimization process. The results

---

[4]https://github.com/releaunifreiburg/QuickTune
[5]https://github.com/releaunifreiburg/DPL
[6]https://github.com/automl/ifBO/tree/48ec25ed7997e653e2c5f4ffbd99eef60590f638
[7]https://github.com/releaunifreiburg/DyHPO
[8]https://github.com/releaunifreiburg/fsbo

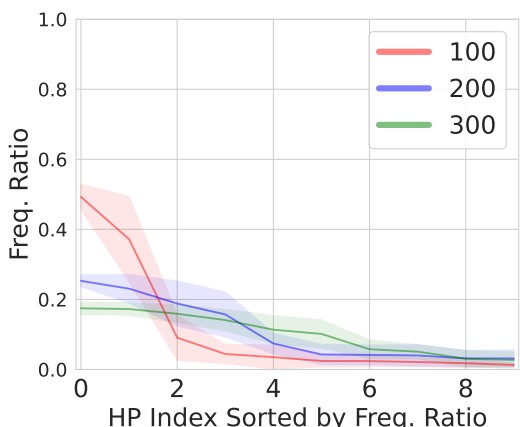

Figure 10: The distribution of top-10 frequently chosen hyperparameter configuration throughout optimization.

clearly indicate that our method explores a wider variety of hyperparameter configurations when the computational budget is large ($B$=300) but focuses on exploiting a smaller subset of configurations when the budget is limited ($B = 100$).

**Visualizations of the normalized regret over BO steps** for LCBench ($\alpha = 4e\text{-}05$), LCBench ($\alpha = 2e\text{-}04$), TaskSet ($\alpha = 4e\text{-}05$), TaskSet ($\alpha = 2e\text{-}04$), PD1 ($\alpha = 4e\text{-}05$), and PD1 ($\alpha = 2e\text{-}04$) are provided Figure 11, 12, 13, 14, 15, and 16, respectively.

**Visualizations of the LC extrapolation over BO steps** for LCBench, TaskSet, and PD1 are provided Figure 17, 18, and 19, respectively. Here, we plot the LC extrapolation results of unseen hyperparameter configurations through BO. Each row shows the results for a different size of the observation set ($|\mathcal{C}| = 0, 10, 50$, and $300$), and each column shows a different size of context points in each LC (0, 2, 5, 10, 20, and 30).

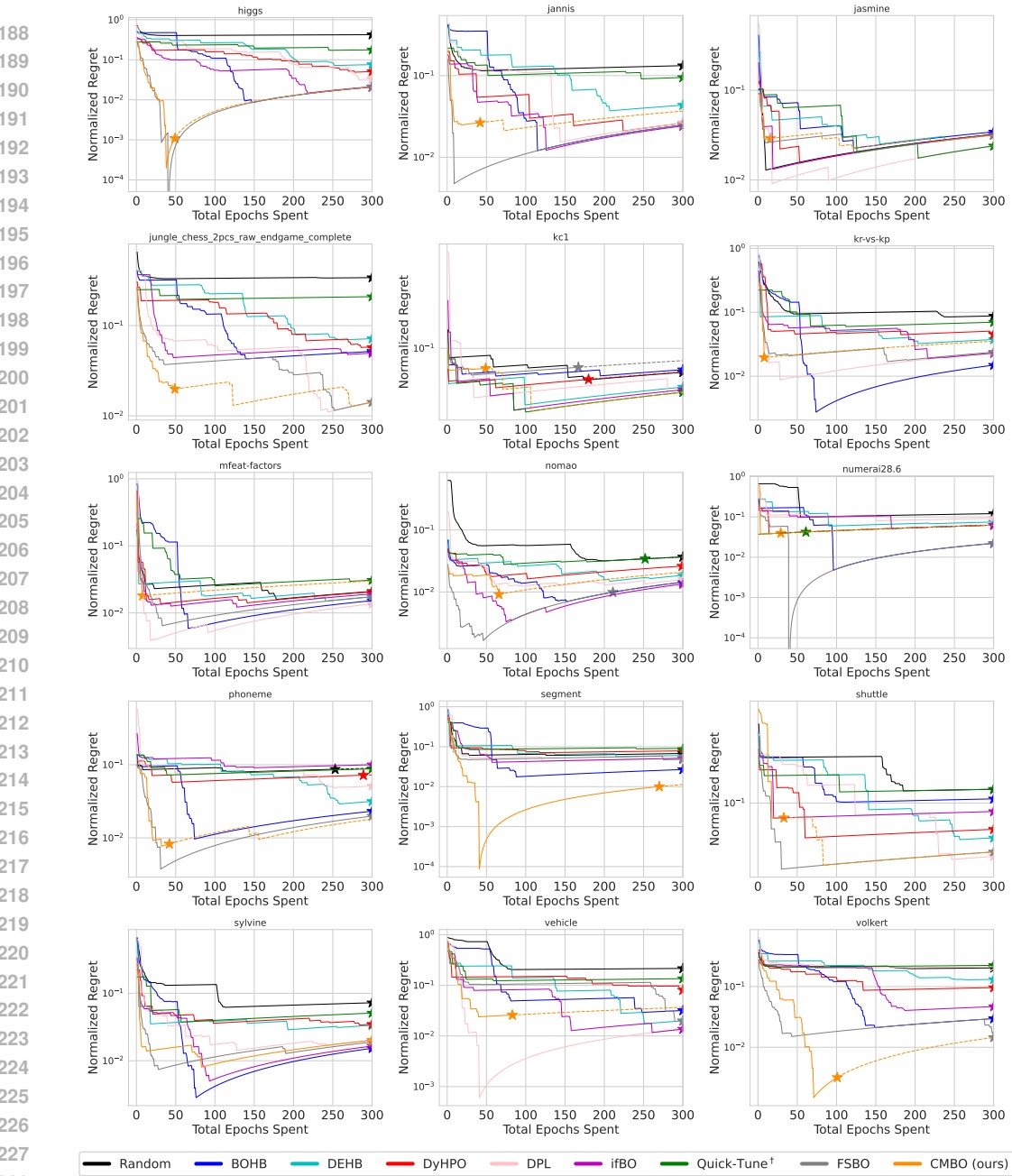

Figure 11: Visualization of the normalized regret over BO steps on **LCBench** ($\alpha =$**4e-05**).

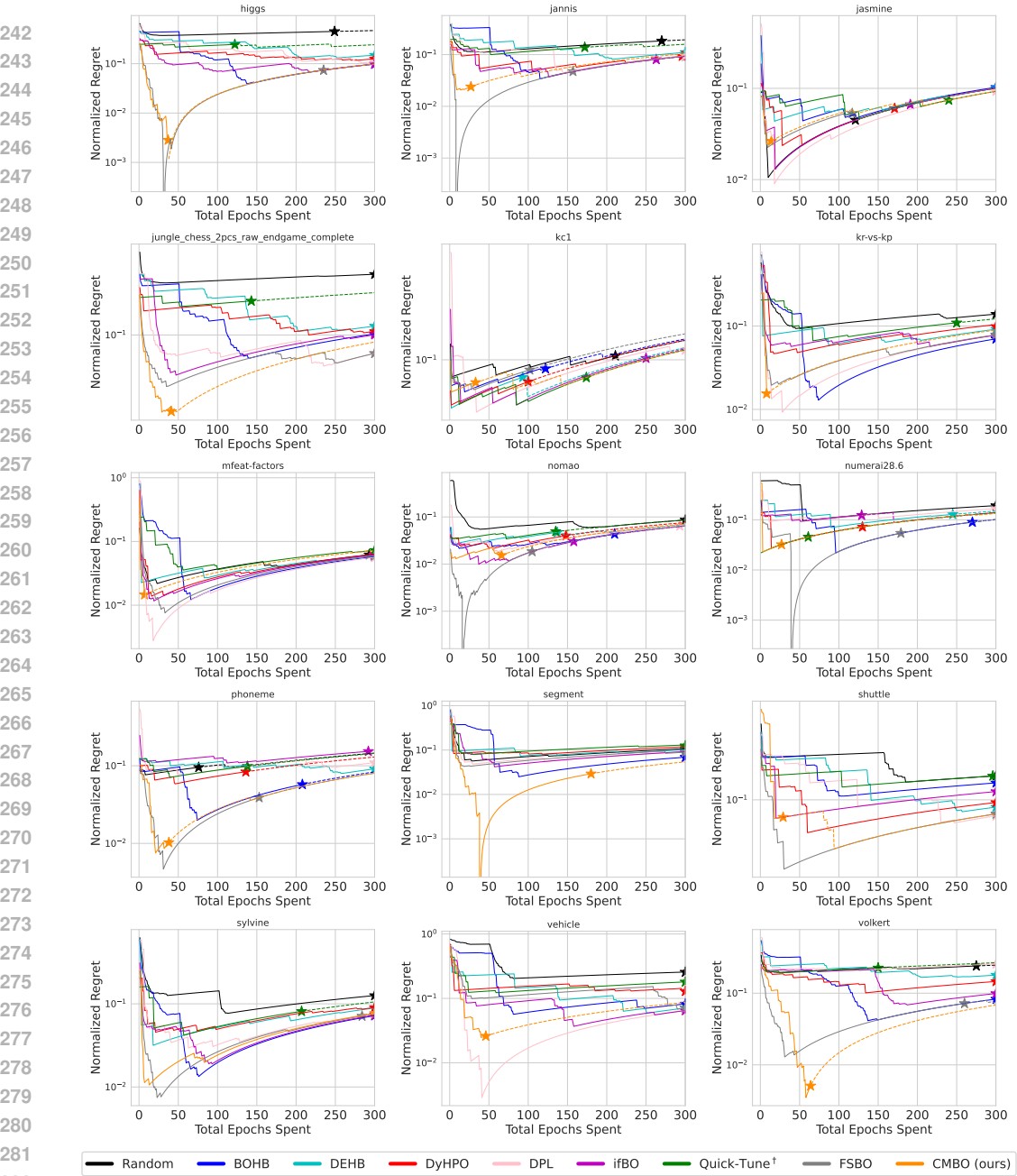

Figure 12: Visualization of the normalized regret over BO steps on **LCBench** ($\alpha$ =**2e-04**).

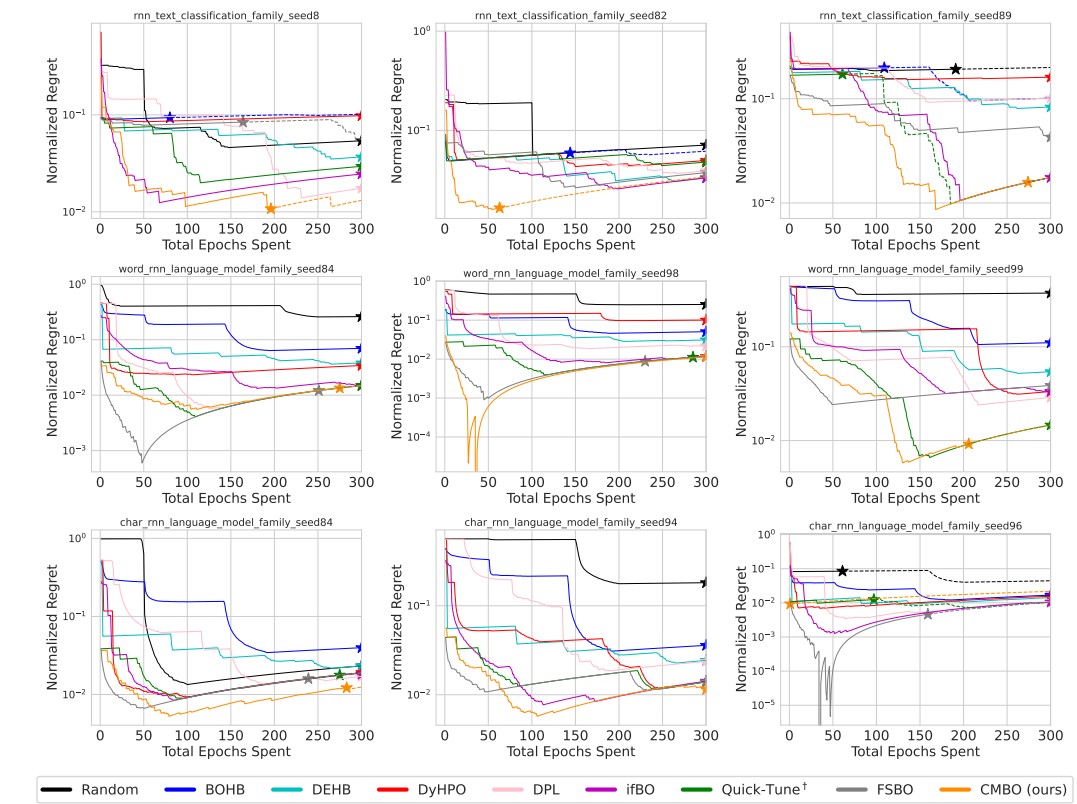

Figure 13: Visualization of the normalized regret over BO steps on **TaskSet** ($\alpha$ =**4e-05**).

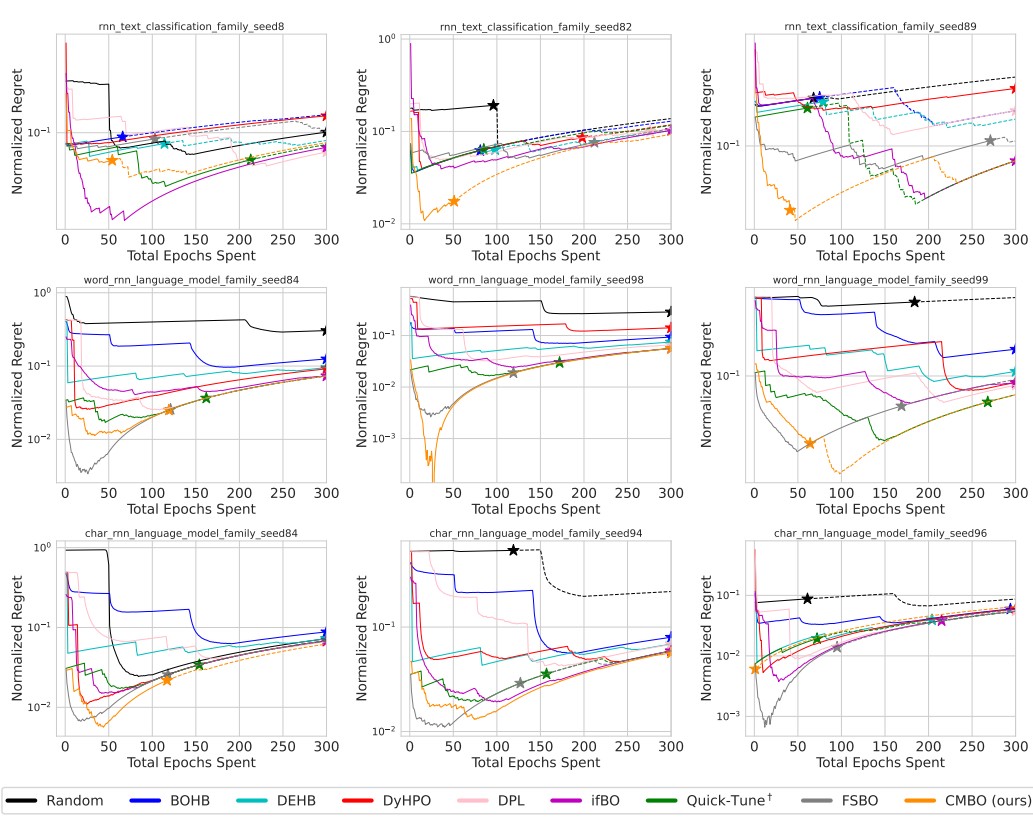

Figure 14: Visualization of the normalized regret over BO steps on **TaskSet** ($\alpha$ =**2e-04**).

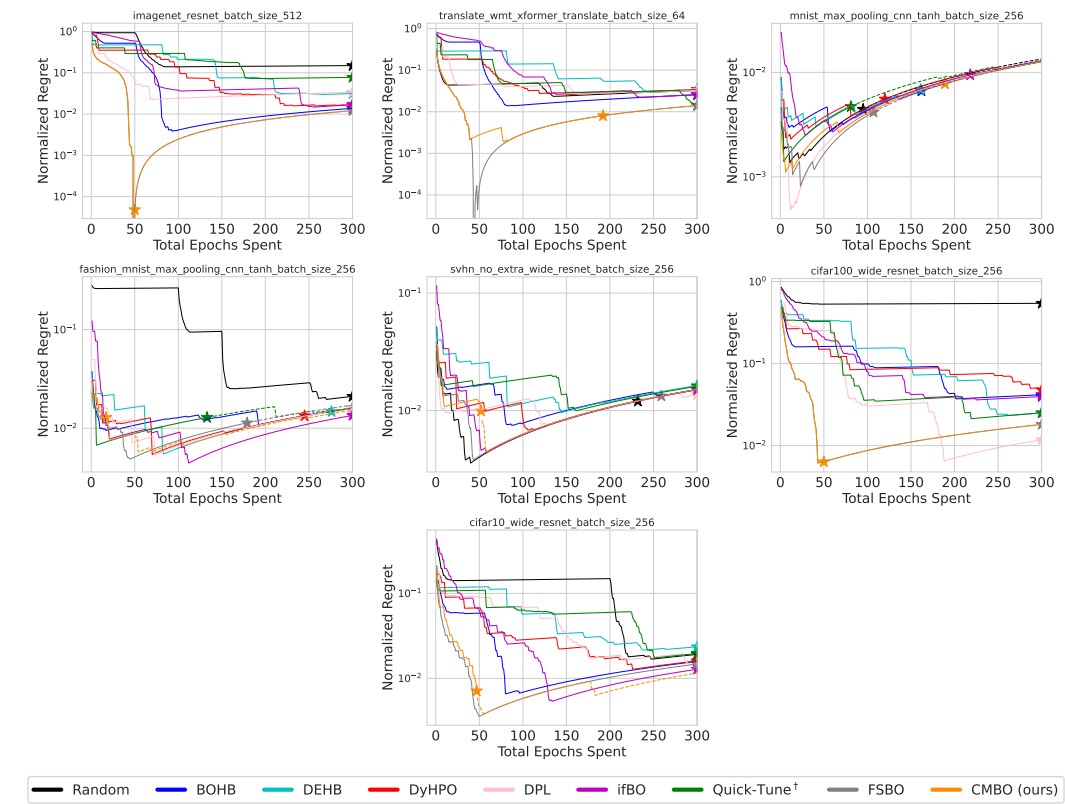

Figure 15: Visualization of the normalized regret over BO steps on **PD1** ($\alpha =$**4e-05**).

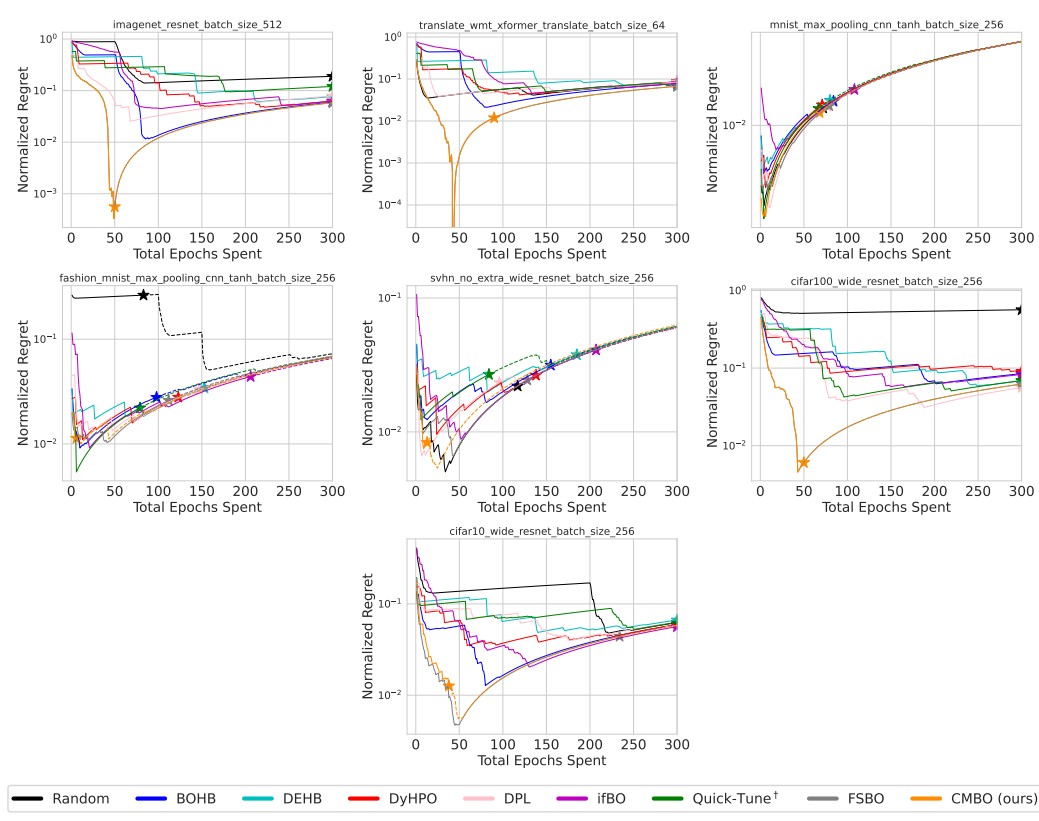

Figure 16: Visualization of the normalized regret over BO steps on **PD1** ($\alpha =$**2e-04**).

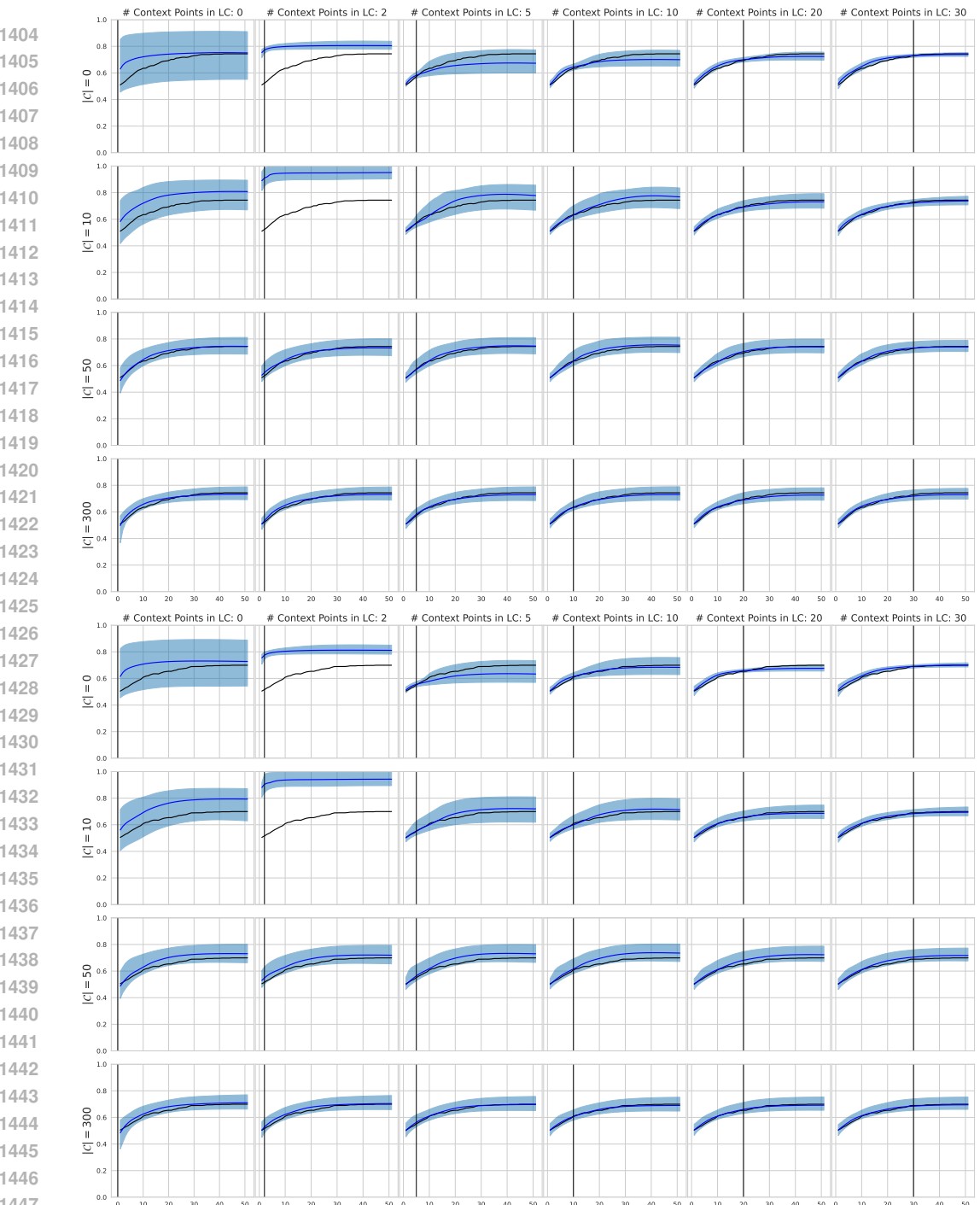

Figure 17: Visualization of LC extrapolation over BO steps on **LCBench**.

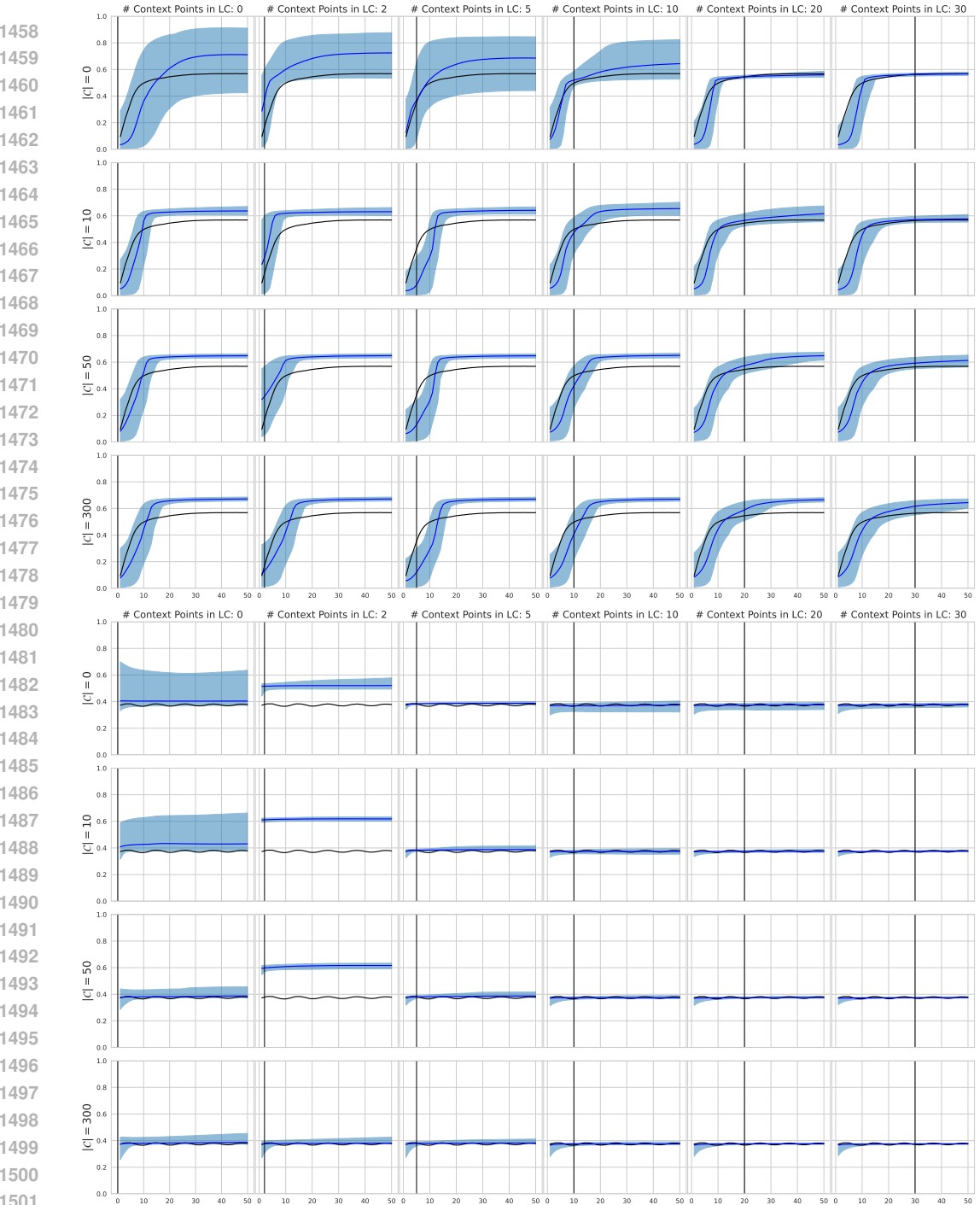

Figure 18: Visualization of LC extrapolation over BO steps on **TaskSet**.

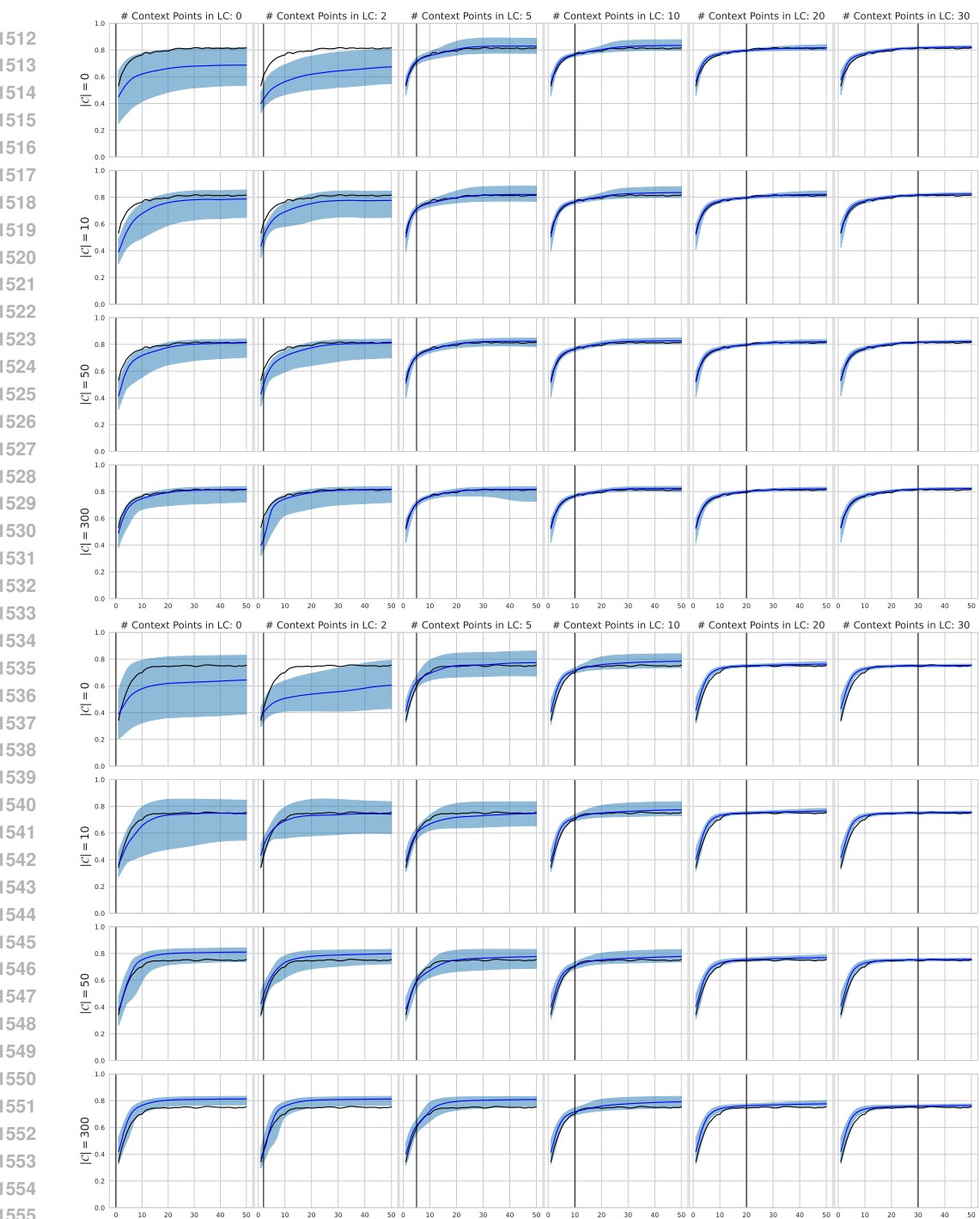

Figure 19: Visualization of LC extrapolation over BO steps on **PD1**.

