# OpenReview forum: "Cost-Sensitive Multi-Fidelity Bayesian Optimization"
_ICLR.cc/2025/Conference — Submitted to ICLR 2025_

### Official Review · Reviewer_fTkx · 2024-10-25

**Soundness:** 3
**Presentation:** 2
**Contribution:** 4
**Rating:** 6
**Confidence:** 5

**Summary:**

A new optimization objective, utility, is proposed in Bayesian optimization. Under this goal, the corresponding acquisition function, multi-fidelity strategy and stop strategy are reconstructed. Among the highlights, utility proposes that it is more suitable for scenarios where users need to balance precision requirements and computing overheads. When a large amount of resources are invested to improve only a small amount of precision, the optimization process can be terminated. As for large and small quantities of quantification, the author proposes a kind of quantitative definition of utility on the user side by allowing users to make choice feedback. Finally, under this index, the method presented in this paper shows excellent advantages.

**Strengths:**

1. The utility is closer to the actual user's consideration, rather than waiting for training convergence, which will waste a lot of computing resources, and the user only gets limited benefits.

2. The utility's quantification design is clever, allowing users to provide data through 2 out of 1 judgment, and then accurately estimate the quantification function. This design is easy to use and more suitable for users.

3. A large number of experiments are conducted under the utility criterion, and the method shows consistent superiority and stability.

**Weaknesses:**

1. Some words are not serious or well-founded, such as One may argue on line 55 and One may argue on line 65, saying that it is difficult for users to obtain the total budget and let users evaluate how to balance benefits and expenses.

2. Figure 2 is called true utility. I think the evaluation here is not true utility. Instead, the author assumes the model and estimates the parameters based on the data generated by the model (user data). However, if the data answered by the user does not match the model set by the author, the fitting effect is much worse.

3. In algorithm 1, if N is given here, how to solve the continuous parameter problem faced by traditional BO? It's supposed to be easy to perfect, but the author didn't do it in this version.

4. This paper explains the ins and outs of utility, but only regret is used as the core index during comparison. If the utility curves, termination points and subsequent trends of different algorithms can be compared, the advantages of the algorithm will be more intuitively understood.

**Questions:**

1. In Equation 2, the expression of b + t, b represents BO step, and t represents training epoch. How do they add up?

2. In algorithm 1, why is t updated in step 12?

3. In Equation 3, utility starts to decrease as bo progresses, so there should be a time when Umax=Uprev, and regret=0, but in the following experiments, there is no time when Umax=Uprev is equal to 0.

4. In equation 5, p_b > 0.5 stops. However, a larger p_b indicates a higher probability of subsequent utility. Why does this stop at this time?

5. Line 448: Multi-fidelity BOs are better than black-box BOs. However, multi-fidelity is not applicable in real industrial scenarios. Since black boxes are used, different epochs cannot be used to terminate the BOs. This statement is not appropriate.

---

> ### Author Response · Authors · 2024-11-22
> **Response to Reviewer fTkx (1/2)**
>
> We sincerely appreciate your time and thoughtful feedback on our work. We have carefully considered your comments and questions and have provided detailed responses below.
>
> ---
>
> **[Q1]** Some words are not serious or well-founded, such as One may argue on line 55 and One may argue on line 65, saying that it is difficult for users to obtain the total budget and let users evaluate how to balance benefits and expenses.
> - Thank you for pointing that out. We have corrected L55 and 64 in the revision.
>
> ---
>
> **[Q2]** Figure 2 is called true utility. I think the evaluation here is not true utility. Instead, the author assumes the model and estimates the parameters based on the data generated by the model (user data). However, if the data answered by the user does not match the model set by the author, the fitting effect is much worse.
> - Thank you for pointing this out. Fig. 2 in its current form is the true utility because we generated the user preference data precisely from this model.
> - However, as you mentioned, in practical scenarios the user preference data will not come from such a predefined model as in Fig. 2. It then becomes inappropriate to call them the true utility, but they will be the “most correct” utility under the given model assumption. There exists the trade-off – more flexible models with more parameters will make the “most correct” utility even more correct, but the required size of user preference data will increase as well.
>
> ---
>
> **[Q3]** In algorithm 1, if N is given here, how to solve the continuous parameter problem faced by traditional BO? It's supposed to be easy to perfect, but the author didn't do it in this version.
> - The pool-based HPO assumption simplifies the problem to align with widely used tabular benchmark datasets (e.g., LCBench, Taskset, PD1). If the pool size becomes too large, the algorithm could indeed be modified to use optimization-based methods (e.g., gradient descent) to directly find the optimal hyperparameter configuration. This approach would slow down the experiments as it requires finding the optimal configuration through inner-optimization at each BO step.
> - While we acknowledge that the discretized hyperparameter assumption is limiting, **this issue extends beyond our paper to the entire multi-fidelity HPO community that currently relies on tabular benchmark datasets for evaluation**(e.g., DyHPO [1], DPL [2], and ifBO [3]). To address this fundamentally, the HPO community needs to propose and adopt new non-tabular benchmark datasets for multi-fidelity HPO.
>
> ---
>
> **[Q4]** This paper explains the ins and outs of utility, but only regret is used as the core index during comparison. If the utility curves, termination points and subsequent trends of different algorithms can be compared, the advantages of the algorithm will be more intuitively understood.
> - The reason we used the normalized regret is to properly average the results over different tasks (e.g. in Fig. 4 and Table 1), following the previous literature.
> - Figure 5 already shows the termination points and subsequent trends of different algorithms.
>
> ---
>
> **[Q5]** In Equation 2, the expression of b + \Delta t, b represents BO step, and \Delta t represents training epoch. How do they add up?
> - In this paper, for notational brevity, we assume that users spend precisely 1 training epoch for each 1 BO step, so their units are the same.
> - Of course, we may assume different amounts of training epochs for each 1 BO step. For instance, if we assume we spend 5 training epochs for each BO step, then the expression will be $5 * b + \Delta t$ (i.e., 5 training epochs have been consumed for each of the last $b$ BO steps + we extrapolate for additional $\Delta t$ training epochs, e.g., 10 epochs)
>
> ---
>
> **[Q6]** In algorithm 1, why is t updated in step 12?
> - The role of $t_n* \leftarrow t_n* + 1$ is to mark that the currently selected hyperparameter configuration $x_n*$ has just been evaluated for one more epoch.
> - This is nothing but from the original freeze-thaw BO – we keep the record of on which epoch each of the hyperparameter configurations is freezed (stopped). This information is used later to dynamically thaw (resume) the selected configuration from the most recently evaluated epoch (i.e., from the up-to-date $t_n*$, i.e., Line 8 in Algorithm 1).
>
> ---
>
> **[Q7]** In Equation 3, utility starts to decrease as bo progresses, so there should be a time when Umax=Uprev, and regret=0, but in the following experiments, there is no time when Umax=Uprev is equal to 0.
> - This seems to be a simple misunderstanding. As utility (= $U_\text{prev}$) starts to decrease, the gap between $U_\text{max}$ and $U_\text{prev}$ will become even greater. Please be cautious of the sign in Eq. (3).
>
> ---

---

> > ### Comment · Reviewer_fTkx · 2024-11-26
> > **Response to the authors**
> >
> > Thanks for the detailed response. Most of the questions are explained well. However, I still have some concerns.
> >
> > In Q2, "most correct" is too subjective. This utility model is proposed by authors with "simple define". Why this simple model can always be suitable for users?
> >
> > In Q7, I am still confused that why Uprev won't be equal to Umax at any time.

---

> ### Author Response · Authors · 2024-11-22
> **Response to Reviewer fTkx (2/2)**
>
> **[Q8]** In equation 5, p_b > 0.5 stops. However, a larger p_b indicates a higher probability of subsequent utility. Why does this stop at this time?
>
> - This is a simple misunderstanding as well. In L277, we explained that "In the former case, we terminate the BO process when $\bf p_b < 0.5$", not when $p_b > 0.5$.
>
> ---
>
> **[Q9]** Line 448: Multi-fidelity BOs are better than black-box BOs. However, multi-fidelity is not applicable in real industrial scenarios. Since black boxes are used, different epochs cannot be used to terminate the BOs. This statement is not appropriate.
> - Thanks for the suggestion. We agreed  and corrected the corresponding part in L416.
>
> ---
>
> **Reference**
>
> [1] Wistuba, Martin, Arlind Kadra, and Josif Grabocka. "Supervising the multi-fidelity race of hyperparameter configurations." Advances in Neural Information Processing Systems 35 (2022): 13470-13484.
>
> [2] Kadra, Arlind, et al. "Scaling laws for hyperparameter optimization." Advances in Neural Information Processing Systems 36 (2023).
>
> [3] Rakotoarison, Herilalaina, et al. "In-Context Freeze-Thaw Bayesian Optimization for Hyperparameter Optimization." arXiv preprint arXiv:2404.16795 (2024).
>
> ---

---

> > ### Author Response · Authors · 2024-11-25
> > **Kind Reminder**
> >
> > Thank you for your dedication and interest in our paper. As the author and reviewer discussion period approaches its end, we are curious to know your thoughts on our rebuttal and whether you have any additional questions.

---

### Official Review · Reviewer_Pvea · 2024-11-03

**Soundness:** 2
**Presentation:** 2
**Contribution:** 2
**Rating:** 5
**Confidence:** 4

**Summary:**

This paper introduces CMBO (Cost-sensitive Multi-fidelity Bayesian Optimization), a novel framework for hyperparameter optimization that explicitly considers the trade-off between computational cost and performance. The key innovation is the introduction of a utility function that captures user preferences regarding this trade-off, which can be learned from preference data. The method combines three main components: (1) a novel acquisition function that maximizes expected utility improvement, (2) an adaptive stopping criterion that determines when to terminate optimization based on utility saturation, and (3) a transfer learning approach using Prior-Fitted Networks (PFNs) with a novel mixup strategy for learning curve extrapolation. The authors evaluate their method on three benchmark datasets (LCBench, TaskSet, PD1) against several baseline methods, demonstrating superior performance especially in scenarios with strong cost penalties.

**Strengths:**

The technical approach is well-developed and carefully constructed. The paper provides detailed explanations of each component, including the mathematical formulations and algorithmic details.

The transfer learning component using PFNs with the proposed mixup strategy shows notable improvement in sample efficiency, which is particularly important for early-stage optimization decisions.

The paper addresses a practical concern in hyperparameter optimization - the need to balance performance gains against computational costs - that is relevant to many real-world applications.

**Weaknesses:**

The paper's fundamental premise regarding the novelty of considering cost-performance trade-offs in multi-fidelity BO is questionable. Many existing multi-fidelity acquisition functions already incorporate such trade-offs implicitly. The authors should better differentiate their approach from existing methods that handle exploration-exploitation balance.

The experimental comparisons may not be entirely fair, as some baselines do not incorporate transfer learning while the proposed method does. A more equitable comparison would include state-of-the-art transfer learning HPO methods.

The paper could benefit from a more thorough ablation study to isolate the contributions of different components, particularly to demonstrate whether the utility function provides benefits beyond what's already captured in traditional multi-fidelity acquisition functions.

Also as a MFBO paper, the paper did not compare to SOTA MFBO methods like MFKG, CFKG, BOCA, MF-UCB.

The claim that existing multi-fidelity BO methods "tend to over-explore" (Line 53) is not well-substantiated and could be contested. The authors should provide empirical evidence for this claim or revise it.

**Questions:**

How does the proposed utility-based acquisition function fundamentally differ from existing multi-fidelity acquisition functions that already balance exploration and exploitation? Could you provide a detailed comparison with specific acquisition functions?

Have you considered comparing your method with non-myopic acquisition functions or RL-based approaches (e.g., work by Hsieh et al. 2021 or Dong et al. 2022) that might address similar concerns about long-term optimization strategy?

Could you provide additional ablation studies that:
- Compare the method without the utility function to isolate its contribution
- Evaluate the performance against other transfer learning HPO methods
- Demonstrate the individual impact of each component (acquisition function, stopping criterion, transfer learning)

The paper assumes users can effectively specify their preferences regarding the cost-performance trade-off. How sensitive is the method to misspecified preferences, and what guidance can be provided to users for setting these preferences effectively?

Could you clarify how the proposed method differs from simply annealing the exploration parameter in traditional acquisition functions? The current presentation makes it difficult to distinguish the novelty of your approach from this simpler alternative.

---

> ### Author Response · Authors · 2024-11-22
> **Response to Review Pvea (1/3)**
>
> We sincerely appreciate your time and thoughtful feedback on our work. We have carefully considered your comments and questions and have provided detailed responses below.
>
> ---
>
> **[Q1]** The paper's fundamental premise regarding the novelty of considering cost-performance trade-offs in multi-fidelity BO is questionable. Many existing multi-fidelity acquisition functions already incorporate such trade-offs implicitly. The authors should better differentiate their approach from existing methods that handle exploration-exploitation balance.
> - Of course the numerous BO acquisition functions have been designed to find the good balance between exploration and exploitation during the optimization. However, what we argue throughout this whole paper is that **they are all insufficient in terms of maximizing utility**. This is simply because the existing acquisition functions are not aware of utility. This has been evidenced by the whole experimental section, especially Figure 7a, 7b, and 7c.
>
> ---
>
> **[Q2]** The experimental comparisons may not be entirely fair, as some baselines do not incorporate transfer learning while the proposed method does. A more equitable comparison would include state-of-the-art transfer learning HPO methods.
> - We have already included state-of-the-art transfer learning baselines such as Quick-Tune$^\dagger$ [1] (i.e., transfer version of DyHPO [2]) and FSBO [3]. The superiority of our method over these baselines, as demonstrated through extensive experiments, sufficiently highlights the efficiency of our approach for cost-sensitive multi-fidelity HPO.
> - It is natural that some of the baselines are not transfer-learning methods, because our model covers not only transfer learning, but also multi-fidelity BO. It is natural to compare against the existing multi-fidelity BO methods which are not necessarily transfer-learning methods.
>
> ---
>
> **[Q3]** The paper could benefit from a more thorough ablation study to isolate the contributions of different components, particularly to demonstrate whether the utility function provides benefits beyond what's already captured in traditional multi-fidelity acquisition functions.
> - Thank you for pointing this out. Please refer to our answer to your **[Q10]**.
>
> ---
>
> **[Q4]** Also as a MFBO paper, the paper did not compare to SOTA MFBO methods like MFKG, CFKG, BOCA, MF-UCB.
> - We have already compared our approach with the most recent state-of-the-art MFBO methods like DPL [3] (NeurIPS 2023) and ifBO [4] (current SOTA, ICML 2024). We strongly believe that the superiority of our method over those baselines described in the whole experimental section is sufficient to show the efficiency of our method for cost-sensitive HPO.
> - We have included the discussion about MFKG, CFKG, BOCA, MF-UCB in Appendix A.
>
> ---
>
> **[Q5]** The claim that existing multi-fidelity BO methods "tend to over-explore" (Line 53) is not well-substantiated and could be contested. The authors should provide empirical evidence for this claim or revise it.
> - We have already provided sufficient empirical evidence like Fig. 7b and 7c as follows:
> - Fig. 7b shows when the performance of configuration chosen by each method is maximized in the future with increment $\Delta t$. We found that baselines choose configurations that have larger  $\Delta t$ than ours over all the optimization process.
> - Fig. 7c shows the distribution of the top-10 most frequently selected configurations during the BO. The top-10 configuration distribution of baseline is much flatter than ours even when the penalty is the strongest.
> ---
>
> **[Q6]** How does the proposed utility-based acquisition function fundamentally differ from existing multi-fidelity acquisition functions that already balance exploration and exploitation? Could you provide a detailed comparison with specific acquisition functions?
> - Please refer to our answer to your **[Q1]**.
>
> ---
>
> **[Q7]** Have you considered comparing your method with non-myopic acquisition functions or RL-based approaches (e.g., work by Hsieh et al. 2021 or Dong et al. 2022) that might address similar concerns about long-term optimization strategy?
> - We have already compared our method with the following non-moypic acquisition functions.
> - DPL [3] proposed to learn power law functions for the learning curve extrapolation. Then, it uses expected improvement (EI) at the maximum budget as an acquisition function, which is non-moypic.
> - ifBO [4] proposed a variant of PFNs that can extrapolate learning curves using observations from partial learning curves of various configurations. Then, it uses probability improvement (PI) at the random future (i.e., $\Delta t \sim \mathbb{U}(0, T-t)$) as an acquisition function, which is non-moypic.
> - The reason our acquisition function is superior to those baselines is because our acquisition can dynamically adjust the degree of being myopic/non-myopic, as clearly explained in L465-482 and Figure 7a, 7b, and 7c.
>
> ---

---

> > ### Comment · Reviewer_Pvea · 2024-11-22
> >
> > Thank you for your detailed response.
> >
> > I am still confused about "they are all insufficient in terms of maximizing utility."
> >
> > How the defined utility is related to optimization metrics (such as simple regret)? Is there any mathematical connection? Why a "good" utility will lead to better regret convergence? Plus, what is a "good" utility in definition?
> >
> > Why \gamma= log2 5 and \beta = e−1 is chosen as utility and why they are representative? What if we use different numbers?

---

> ### Author Response · Authors · 2024-11-22
> **Response to Review Pvea (2/3)**
>
> **[Q8]** Compare the method without the utility function to isolate its contribution
> - The conventional MFBO setting without utility is exactly the same as when we set $\alpha=0$.
> - We have already discussed our contribution in this conventional MFBO setting in the L425-448.
> ---
> **[Q9]** Evaluate the performance against other transfer learning HPO methods
> - Please refer to our answer to your **[Q2]**.
> ---
> **[Q10]** Demonstrate the individual impact of each component (acquisition function, stopping criterion, transfer learning)
> - Thank you for pointing this out. We conducted ablation studies on our stopping criterion ($p_b$), acquisition function (Acq.), and transfer learning (T.) with mixup strategy on the PD1 benchmark.
> - For the stopping criterion, we either use the smoothly-mixed criterion with $\beta=-1$ as in our full method ($p_b$ ✅), or use the regret-based criterion with $\beta\rightarrow 0$, the one used by the baselines ($p_b$ ❌).
> - For the acquisition function, we either use Eq.(2) (Acq. ✅) or the acquisition function of ifBO [5] (Acq. ❌).
> - For transfer learning, we either use our surrogate trained with the proposed mixup strategy (T. ✅) or the surrogate of ifBO [5] (T. ❌).
> - The results in the below table show that the performance improves sequentially as each component is added, with more pronounced improvements under strong penalties ($\alpha$ = 2e-4).
> - Notably, the stopping criterion does not affect the results in the conventional setting ($\alpha$ = 0).
> - We have included these results in Table 3 in the revision.
>
> | **$p_b$**                    | **Acq.**                | **T.**                 | **$\alpha=0$**        | **$\alpha=4e\text{-}05$** | **$\alpha=2e\text{-}04$** |
> |------------------------------|-------------------------|------------------------|-----------------------|---------------------------|---------------------------|
> | ❌                           | ❌                      | ❌                     | 0.8 ± 0.1            | 2.0 ± 0.1                 | 5.8 ± 0.6                 |
> | ❌                           | ❌                      | ✅                     | **0.2 ± 0.0**        | 1.4 ± 0.0                 | 5.7 ± 0.3                 |
> | ❌                           | ✅                      | ✅                     | **0.2 ± 0.0**        | 1.2 ± 0.0                 | 4.4 ± 0.0                 |
> | ✅                           | ✅                      | ✅                     | **0.2 ± 0.0**        | **0.8 ± 0.0**            | **0.9 ± 0.0**            |
>
> ---
> **[Q11]** The paper assumes users can effectively specify their preferences regarding the cost-performance trade-off. How sensitive is the method to misspecified preferences, and what guidance can be provided to users for setting these preferences effectively?
> - This is nothing but the general noisy label problem. We believe that we do not need to specifically address this problem in this paper, as it is out of the scope of this paper.
> ---
> **[Q12]** Could you clarify how the proposed method differs from simply annealing the exploration parameter in traditional acquisition functions? The current presentation makes it difficult to distinguish the novelty of your approach from this simpler alternative.
> - Simply annealing the extrapolation parameter in traditional acquisition functions (e.g, annealing the coefficient multiplied with standard deviation in UCB) might slightly improve the performance, but it doesn’t change the fundamental fact that those acquisition functions are not aware of user utility. Therefore, such simple annealing cannot be an alternative to our acquisition function which directly aims to maximize the utility.
> ---

---

> ### Author Response · Authors · 2024-11-22
> **Response to Review Pvea (3/3)**
>
> **Reference**
>
> [1] Arango, Sebastian Pineda, et al. "Quick-tune: Quickly learning which pretrained model to finetune and how." arXiv preprint arXiv:2306.03828 (2023).
>
> [2] Wistuba, Martin, Arlind Kadra, and Josif Grabocka. "Supervising the multi-fidelity race of hyperparameter configurations." Advances in Neural Information Processing Systems 35 (2022): 13470-13484.
>
> [3] Wistuba, Martin, and Josif Grabocka. "Few-shot bayesian optimization with deep kernel surrogates." arXiv preprint arXiv:2101.07667 (2021).
>
> [4] Kadra, Arlind, et al. "Scaling laws for hyperparameter optimization." Advances in Neural Information Processing Systems 36 (2023).
>
> [5] Rakotoarison, Herilalaina, et al. "In-Context Freeze-Thaw Bayesian Optimization for Hyperparameter Optimization." arXiv preprint arXiv:2404.16795 (2024).
>
> ---

---

> ### Author Response · Authors · 2024-11-22
> **Response to the comment**
>
> Thank you for your quick reply.  We respond your questions as follows:
>
> ---
>
> **[Q1]** How the defined utility is related to optimization metrics (such as simple regret)? Is there any mathematical connection? Why a "good" utility will lead to better regret convergence? Plus, what is a "good" utility in definition?
>
> ---
>
> - In this paper, utility is just the target we want to maximize by definition. Utility is not something that can lead to better optimization convergence. It is just the target. There is no "good" or "bad" utility. Utility is just given by each user, according to the user's preference about the trade-off between the cost and performance of BO. In this sense, the acquisition functions of previous BO are all insufficient because they are not aware of the target in the first place!
>
> ---
>
> **[Q2]** Why \gamma= log2 5 and \beta = e−1 is chosen as utility and why they are representative? What if we use different numbers?
>
> - $\gamma$ and $\beta$ are related to the stopping criterion, not the definition of the utility. This is clearly explained in L243-274 of the manuscript.
>
> - We chose $\gamma = \log_2 5$ because it results in $\delta_b = 0.2$ when $p_b = 0.5$, as shown in Fig. 3. This means that when the model is uncertain about improvement ($p_b = 0.5$), our stopping criterion aligns with the baselines that use a fixed $\delta_b = 0.2$.
>
> - $\beta$ is a hyperparameter in our method that adjusts the threshold $\delta_b$ based on the probability of improvement $p_b$. If the model is highly confident about improvement ($p_b \rightarrow 1$), then $\delta_b \rightarrow 1$, and the stopping criterion in Eq. (3) is never satisfied. Conversely, if the model is certain there will be no improvement ($p_b \rightarrow 0$), then $\delta_b \rightarrow 0$, and the stopping criterion in Eq. (3) is always satisfied.
>
> - We have already discussed the effect of the hyperparameter $\beta$ in our algorithm in Fig. 7d.

---

> > ### Author Response · Authors · 2024-11-25
> > **Kind Reminder**
> >
> > Thank you for your dedication and interest in our paper. As the author and reviewer discussion period approaches its end, we are curious to know your thoughts on our rebuttal and whether you have any additional questions.

---

> > ### Comment · Reviewer_Pvea · 2024-11-26
> >
> > Thank you for your quick response. I have some concerns and would appreciate further clarification:
> >
> > I am still unconvinced about how introducing this utility function and maximizing it provides practical benefits for the optimization task. My understanding is that Bayesian Optimization (BO) is typically employed because black-box optimization is too complex for designers to understand or perform manually. BO serves as a remedy for this complexity. Ultimately, isn’t the goal to find the optimal solution with minimal resources? How the defined utility can benefit us? And most importantly, under what definition it can help us?
> >
> > Also, if the utility does not directly influence optimization convergence, could you elaborate further on how it meaningfully complements the optimization process? Moreover, while acquisition functions in prior BO approaches may not explicitly target user-defined preferences, they still inherently drive optimization toward regret minimization. How does your utility-driven approach reconcile or improve on this?
> >
> > In summary, while I appreciate the insights shared, I still find the connection between the utility-driven approach and practical optimization outcomes unclear. Could you provide more intuition or examples demonstrating how this method outperforms traditional BO approaches in scenarios with real-world constraints?

---

> > > ### Comment · Reviewer_Pvea · 2024-11-30
> > >
> > > Dear Authors,
> > >
> > > Could you please respond to my previous question?
> > >
> > > At this stage, I am not convinced how the proposed method can be useful in practice. Please see the detailed question in the previous comments. I am open to increasing my rating provided my major concern is resolved.

---

### Official Review · Reviewer_TsBW · 2024-11-03

**Soundness:** 3
**Presentation:** 1
**Contribution:** 2
**Rating:** 5
**Confidence:** 3

**Summary:**

This work developed a new framework of multi-fidelity Bayesian optimization in consideration of budget/cost penalty. An improved acquisition function, as a variant of the expected improvement (EI), was proposed to represent the cost-performance utility of decision makers. The surrogate models were built by the learning curve extrapolation method in conjunction with a data augmentation strategy. Additionally, a stopping criterion was introduced to adaptively save the cost, thereby achieving best utility. Finally, the HPO experiments demonstrated the effectiveness and superiority of the framework.

**Strengths:**

1. The motivations are clear. Existing multi-fidelity optimization methods mainly assumed a given budget, little focusing on the active learning strategy with limited cost.
2. The solutions are innovative, including the improved acquisition function and the data augmentation strategy in transfer learning.
3. The experiments is well-organized to show the effectiveness and implications.

**Weaknesses:**

1. The presentation is not good enough. The utility function is hard to define and learn from noisy preference data. As aligning utility from preference learning was not performed in the experiments, a definition or assumption should be enough rather than trivial content from line 182 to line 201. On the other hand, the basic definition of multi-fidelity optimization problems were missing. It is not mentioned whether it is a maximization or minimization problem. This should be highly related to the design of utility function.
2. The solution, especially the improved acquisition function, may not work in cost-limited problems. This work only introduces the penalty rather than constraints. In other word, if there were cost constraints, this framework may not ensure the cost during search is within the constraints.
3. The experiment settings seem unfair. The metric was related to utility $U$, however, it is not clear whether other baselines considered $U$ as their objectives as well. If not, the comparison may be unfair due to different objectives.

**Questions:**

1. How to determine the hyperparameters in the framework, such as $\delta_b$ in the stopping criterion?
2. In line 412, why use different $\beta$ for the proposed method only?
3. Why not consider a bigger $\alpha$ and other format of cost independent of the budget number $b$?

---

> ### Author Response · Authors · 2024-11-22
> **Response to Reviewer TsBW (1/2)**
>
> We sincerely appreciate your time and thoughtful feedback on our work. We have carefully considered your comments and questions and have provided detailed responses below.
>
> ---
> **[Q1]** The presentation is not good enough. The utility function is hard to define and learn from noisy preference data. As aligning utility from preference learning was not performed in the experiments, a definition or assumption should be enough rather than trivial content from line 182 to line 201.
> - In the rebuttal period, we considered the following scenario for learning the utility function from user preference data.
> - We assume that the user wants to set the trade-off (between the cost and performance of BO) achievable by running other multi-fidelity HPO methods, such as ifBO. We run ifBO on all the PD1 meta-training tasks and average those BO trajectories, obtaining a single BO trajectory corresponding to the overall representative trade-off. Based on that single curve, we randomly sample many points around that curve, such that all the points locate either upper or bottom parts of the curve. Then, we can collect infinitely many pairs of points by randomly picking one upper point and one bottom point (with the constraint that the upper one should have larger budgets than the bottom one), and construct the user preference data. We then estimated the utility function and conducted experiments using this utility. The estimated utility and BO trajectory is depicted in Fig. 9 (please see Appendix C or page 18 in the revision).
> - The table below shows that our method consistently outperforms all the baselines we consider. We have included the results in the revision (the first column in Table 2).
> Method | Random            | BOHB             | DEHB              | DyHPO             | DPL               | ifBO              | QuickTune$^\dagger$ | FSBO             | CMBO (ours)      |
> |--------|-------------------|------------------|-------------------|-------------------|-------------------|-------------------|---------------------|------------------|------------------|
> | Regret | 25.5$\pm$7.7 | 9.2$\pm$0.9 | 14.8$\pm$6.3 | 10.9$\pm$1.6 | 15.5$\pm$7.0 | 11.2$\pm$1.5 | 19.5$\pm$0.0   | 7.4$\pm$0.0 | **2.1$\pm$0.0** |
> | Rank   | 6.9               | 5.1              | 6.4               | 5.9               | 5.9               | 6.3               | 4.8                 | 2.8              | **1.0**              |
> ---
> **[Q2]** On the other hand, the basic definition of multi-fidelity optimization problems were missing. It is not mentioned whether it is a maximization or minimization problem. This should be highly related to the design of utility function.
> - Thank you for pointing this out. We have included it in L161.
> ---
> **[Q3]** The solution, especially the improved acquisition function, may not work in cost-limited problems. This work only introduces the penalty rather than constraints. In other words, if there were cost constraints, this framework may not ensure the cost during search is within the constraints.
> - This is a misunderstanding. Our utility function is trivially generalizable to hard constraints, such as maximum BO budgets $B$. It is done by setting the value of utility to $-\infty$ when $b > B$.
> - We empirically verify it by setting $B \in$ {$100, 200, 300$} and plot the same figure as Figure 7c. Figure 10 in Appendix $\S$H shows the results, and we can see that as the total budget $B$ gets smaller, the BO tends to stick to only a few configurations during the BO for more exploitation, as expected.
> ---
> **[Q4]** The experiment settings seem unfair. The metric was related to utility U, however, it is not clear whether other baselines considered U as their objectives as well. If not, the comparison may be unfair due to different objectives.
> - Introducing the concept of utility and addressing how to carefully consider it during the BO are our contributions. Modifying the objective of the baselines will result in very different methods than the original baselines.
> - At least we allow the baselines to early-stop the BO with the regret-based stopping criterion, as clearly explained in L373-375. We believe that the comparisons are meaningful and fair enough.
> ---
> **[Q5]** How to determine the hyperparameters in the framework, such as δ_b in the stopping criterion?
> - For our method, $\delta_b$ changes w.r.t the probability of improvement ($p_b$) , which is clearly explained in L257-274.
> - For baselines, $\delta_b$ is a fixed hyperparameter, therefore, we chose the best-performing $\delta_b= 0.2$ for the baselines using the meta-training datasets.
> - The same value, $\delta_b=0.2$ of baselines was then applied to our method (corresponds to $\gamma=log_2 5$) for a fair comparison.
> ---

---

> > ### Comment · Reviewer_TsBW · 2024-11-26
> > **Response to the authors**
> >
> > I appreciate the authors' effort for response. My concerns still remain regarding the contributions, i.e., how the community can benefit from this work. In all, the paper address the problem for balancing between conducting BO evaluations and stopping early according to user's preference. These two objectives are negatively correlated. However, the authors proposed to aggregate them into a single-objective optimization problem (Equation 2). Different from pursuing the Pareto front as in multi-objective optimization, I think the solution obtained in this work using aggregation may not be appreciated by the researchers of practitioners.
> > 1. Consider the one-step optimization, which is very popular in off-line optimization, is the obtained solution from EI with large penalty terms competitive with that tailored in off-line optimization?
> > 2. Is there any chance that this work could be extended in theory, such as the regret of optimization or the upper bound of cumulated cost?
> > 3. The stopping criteria is totally designed from heuristics. What will happen if there is no specific stopping criteria? The work only considers total evaluation budget as the cost, which is somewhat weired to me since BO (or EI in this work) is already a principled framework for near-optimal data-efficient search, and is more principled. To me, the main contributions in this work are more like considering a different black-box function to be optimized instead of an improvement of existing method like EI.

---

> ### Author Response · Authors · 2024-11-22
> **Response to Reviewer TsBW (2/2)**
>
> **[Q6]** In line 412, why use different β for the proposed method only?
> - This is because $\beta$ is a hyperparameter specific to our method. As explained in the footnote on page 6 (L321-323), the PI criterion in Eq. (5) which uses $\beta$ is based on our novel acquisition function with utility. In contrast, the baselines rely solely on the regret-based criterion in Eq. (3), which does not involve $\beta$.
> - We have removed L374 in the revision to avoid the confusion.
> ---
>
> **[Q7]** Why not consider a bigger α and other format of cost independent of the budget number b?
> - We have already shown the results by considering a bigger $\alpha$ (e.g., 0.00459) using various utility functions  in Table 2.
> - We have already shown the results on cost independent of the budget number $b$ in Fig. 4 by setting $\alpha=0$ which is just the conventional HPO setting.
> ---

---

> > ### Author Response · Authors · 2024-11-25
> > **Kind Reminder**
> >
> > Thank you for your dedication and interest in our paper. As the author and reviewer discussion period approaches its end, we are curious to know your thoughts on our rebuttal and whether you have any additional questions.

---

### Official Review · Reviewer_wRPD · 2024-11-05

**Soundness:** 2
**Presentation:** 3
**Contribution:** 3
**Rating:** 5
**Confidence:** 3

**Summary:**

The paper proposes CMBO, a cost-sensitive multi-fidelity BO methods, which target hyper-parameter tuning problems. CMBO is built upon freeze-thaw BO, which allows for pausing (freeze) the configuration run at intermediate epoch, and resuming (thaw) the run at with remaining epoch later. Specifically, CMBO introduces the utility function to account for the trade-off between the cost and performance of BO steps, then propose a EI-based acquisition function for this utility. CMBO’s optimization policy is based on maximizing the utility (the expectation of the proposed utility), instead of maximizing the validation performance as previous methods. Moreover, to support these steps, CMBO adopts Prior-fitted Network concept to extrapolate the learning curves (LC), which allows for computing the utility-based acquisition function.

**Strengths:**

-	The idea of extrapolating LC to compute the acquisition function (AF) sounds interesting to me.
-	The authors also improve the problem of PFN cannot be trained with dataset, and instead need prior distribution. The mixup strategy fixes this issue.
-	There are many analyses to support the results.
-	The ranking results in Tables 1, 3, 4 seem to be impressive.

**Weaknesses:**

-	It’s strange that the reported results are in terms of normalized regret of the utility U, which seems to be not common in the field. The utility is the term proposed by the authors. FSBO, ifBO and QuickTune seem to use the normalized regret of the function evaluation f(x).

**Questions:**

-	In Table 1, the results of baseline methods change when alpha changes. Is this because of the different normalized regret that I mentioned in the Weakness section? This is because alpha is a parameter of utility function, which only belongs to the proposed CMBO. Other methods, such as Random Search should not be affected by this parameter. Can the authors provide addition results - the normalized regret of evaluation values f(x) - as other baselines?

-	Are PFNs trained only once, or do we need to retrain PFN during CMBO? How does the training time of PFNs compare to the evaluation of a HPO epoch?

-	Minor: Can the authors explain more about the choice of PFNs? How about using Deep Gaussian Process as FSBO baseline?

---

> ### Author Response · Authors · 2024-11-22
> **Response to Reviewer wRPD**
>
> We sincerely appreciate your time and thoughtful feedback on our work. We have carefully considered your comments and questions and have provided detailed responses below.
>
> ---
>
> **[Q1]** It’s strange that the reported results are in terms of normalized regret of the utility U, which seems to be not common in the field. The utility is the term proposed by the authors. FSBO, ifBO and QuickTune seem to use the normalized regret of the function evaluation f(x).
>
>
> - The whole message of this paper is that we need to consider utility as the primary metric. Therefore, it is natural that we report results in terms of utility, including all the baselines and our method.
>
>
> - The reason it is not common in the field is because it is newly introduced in this work, which means that formalizing the concept of utility is our contribution and novelty.
>
> ---
>
> **[Q2]** In Table 1, the results of baseline methods change when alpha changes. Is this because of the different normalized regret that I mentioned in the Weakness section? This is because alpha is a parameter of utility function, which only belongs to the proposed CMBO. Other methods, such as Random Search should not be affected by this parameter. Can the authors provide additional results - the normalized regret of evaluation values f(x) - as other baselines?
> - In this paper, (the normalized regret of) utility is the evaluation metric, which is why the performance of all the baselines and our method changes as $\alpha$ changes.
> - For example, if we use F1-score instead of accuracy for binary classification, the performance value will change naturally. Similarly, a change in $\alpha$ modifies the utility function, changing the performance values.
>
> ---
>
> **[Q3]** Are PFNs trained only once, or do we need to retrain PFN during CMBO? How does the training time of PFNs compare to the evaluation of a HPO epoch?
> - Yes, PFNs are trained only once, and they are used as an in-context inference machine during the BO. We do not need to retrain them at all during the BO.
> - Since no additional training is done during the BO, we only need to consider the inference time of PFNs. It is very marginal compared to other learning curve extrapolation methods, such as DyHPO [1] and DPL [2], which requires retraining throughout the BO.
>
> ---
>
> **[Q4]** Minor: Can the authors explain more about the choice of PFNs? How about using Deep Gaussian Process as FSBO baseline?
> - This is actually what we tried initially. The main problem of using Deep GP for multi-fidelity BO is twofold.
> - First, unlike the black-box method such as FSBO, in multi-fidelity BO the size of input is much larger because the GP kernel needs to model the entire learning curve, not the last validation performances. Therefore, the complexity of GP sharply increases as we collect more observations. We tried several approximations, such as SVGP [3], but was not able to find a good balance between the quality of approximation and the computational cost.
> - Second, it is quite difficult to choose the suitable kernel that can stably model the correlations between different points in a learning curve. We already tried it, but the simple RBF or Matern kernel completely failed. The situation is different from FSBO because it is black-box hence needs not consider the dynamics of learning curves.
> - For the reasons above, we strongly recommend using PFNs as an off-the-shelf learning curve extrapolator.
>
> ---
>
> **Reference**
>
> [1] Wistuba, Martin, Arlind Kadra, and Josif Grabocka. "Supervising the multi-fidelity race of hyperparameter configurations." Advances in Neural Information Processing Systems 35 (2022): 13470-13484.
>
> [2] Kadra, Arlind, et al. "Scaling laws for hyperparameter optimization." Advances in Neural Information Processing Systems 36 (2023).
>
> [3] Hensman, James, Alexander Matthews, and Zoubin Ghahramani. "Scalable variational Gaussian process classification." Artificial Intelligence and Statistics. PMLR, 2015.
>
> ---

---

> > ### Author Response · Authors · 2024-11-25
> > **Kind Reminder**
> >
> > Thank you for your dedication and interest in our paper. As the author and reviewer discussion period approaches its end, we are curious to know your thoughts on our rebuttal and whether you have any additional questions.

---

### Official Review · Reviewer_8PmF · 2024-11-07

**Soundness:** 2
**Presentation:** 1
**Contribution:** 3
**Rating:** 3
**Confidence:** 5

**Summary:**

This manuscript seeks to address the problem of hyperparameter optimization particularly for the training of machine learning models which routinely provide low fidelity performance signals that can be leveraged to improve the efficiency of the optimization procedure. More specifically, this paper proposes numerous improvements to multi-fidelity Bayesian optimization (BO), including (i) a generalized notion of performance that accounts for cost and (ii) an acquisition function based on the expected improvement in the full trajectory of extrapolated future performance outcomes (rather than the typical one-step ahead outcome). The generalized performance, which the authors call "utility", can be specified analytically/parametrically, or otherwise learned from a user's preference. The authors further propose a cost-based stopping criterion for the BO procedure according to values of this "utility". Finally, the authors investigate the use of in-context learning frameworks such as prior-fitted networks (PFNs) to accurately extrapolate learning curves in a few-shot manner by transfer learning from curves of hyperparameter configurations from related tasks/datasets.

**Strengths:**

## Significance

This manuscripts seeks to simultaneously improves on various aspects of BO, including multi-task BO (warm-starting BO from related tasks), multi-fidelity BO (leveraging low fidelity performance signals), cost-aware BO, and also optimal stopping for BO, all in a coherent and unified manner. These aspects are timely and important long-standing open problems in BO. In spite of some potential shortcomings (identified below), some of the high-level ideas and concepts could readily translate to other multi-fidelity frameworks of this kind.

## Originality

This work proposes novelties in number of different components, e.g. in the acquisition function by a) generalizing the performance ($y$) to a quantity that is normalized/penalized by budget/cost (which they call a "utility" function), and also b) generalize this quantity to be based on extrapolated performances. Finally, c) to be able to accurately and efficiently carry out this extrapolation in a few-shot manner, they incorporate recent in-context learning approaches based on PFNs, which are trained on priors implicitly specified by benchmark datasets containing learning curve, and adapt augmentation strategies like mix-up to learning curve data to mitigate the possibility of overfitting.

**Weaknesses:**

## Quality

The technical quality of the proposed methodology could be improved. In particular, I found many aspects of the approach to be arbitrary and not well-motivated. For instance, for the stopping criterion described starting *line 250*, the choice of using the BetaCDF with parameters $\beta, \gamma > 0$ and probability $p$ as the probability of improvement (PI), beyond working fine empirically on the benchmark problems considered, seem highly convoluted and totally arbitrary to me.

A major concern I have in the empirical evaluation of the proposed method is in the "normalized regret of utility" (Eq. 3), which is the primary metric that is reported. Beyond being quite complicated to compute (evidenced by lines 417-420), it is also not obvious to me that this is the "holy grail" metric we should be aiming for in the first place. Does this metric not differ depending on the surrogate/extrapolation model of choice? Furthermore, I am unclear as to how this metric is even defined for other methods such as Random, BOHB, etc. which don't explicitly model the performance $y$, and in which it's unclear how the "utility" can be incorporated? I would be interested in seeing a more conventional plot showing the current best performance (or regret) along the vertical axis.

Another concern is that the reported empirical results all display the BO iteration along the horizontal axis, which is highly misleading in the context of multi-fidelity BO. It seems to me that the notion of a BO step means totally different things in different frameworks. For instance, in BOHB, a BO step signifies training a model with a particular hyperparameter configuration to full completion, but in most cases they are trained for a fraction, e.g. 27/81 epochs, resulting in a *fractional* BO step (in this example 1/3rd of a BO step). In contrast, under the proposed framework, a BO step is the advancement of a configuration by a single epoch. Therefore, I am doubtful that the results presented show an apples-to-apples comparison.

Furthermore, in multi-fidelity BO just showing BO steps along the horizontal axis (fractional or otherwise) is not entirely informative either, especially when cost is of interest. In addition to fractional BO steps, I would like to see a plot with the wall-clock time along the horizontal axis.

## Clarity

A significant weakness of this paper is its lack of clarity, particularly in many parts of the paper (describing important technical details) which I found cryptic and difficult to parse. Some examples include:
- *lines 215-219* (details on learning curve extrapolation and how its used to compute the "BO performance")
- *lines 417-420* (details on computing bounds on the "utility") - this is indecipherable

More generally, this manuscript could benefit from more careful copy-editing. Some specific examples (non-exhaustive) of where writing quality could be improved are enumerated in the "Miscellaneous Remarks" section.


### Miscellaneous Remarks

- The overloaded use of the term "utility" is confusing as utility functions already plays a central role in Bayesian decision theory (of which Bayesian optimization is an special case [Garnett, 2023]). As such, statements such as "We call this trade-off utility" (*line 67*), "We introduce the concept of utility, ..." (*line 110*), and "We first introduce the detailed notion of utility function" (*lines 95-96*), are likely to raise eyebrows.
- *line 40* - "receives more attention"
- *line 77* - "hyperparamter"
- *line 78* - "improve it in future"
- *lines 84-86* - ?
- *line 98* - "a recently introduced"
- *line 107* - "a reasonable and stable way" -- "reasonable" the reader can probably infer but what makes a "utility" function "stable"?
- *line 77* - "hyperparmater"
- *line 161* - "surrogate function" -> "surrogate functions"
- *line 199-200* - the sign of the second term in the binary cross-entropy loss is wrong
- *line 431* - "despite of the transfer learning"
- *line 351* - "For easier transfer learning"
- *line 364* - "training epochs at future"
- *line 538* - "numerous empirical evidences"

**Questions:**

- *line 164* - The proposed method works with a fixed, finite pool of hyperparameters. Firstly, I would contend with the claim that this is the "convention" in BO, where it is arguable the exception rather than the rule. However, my biggest question is how this pool is populated in the first place? I would guess randomly, which begs the question of how comparisons are carried out against other methods in which this is not a common practice, e.g. BOHB? Furthermore, details are missing as to how many hyperparameter configurations there are in this pool. Fig 7c hints that this is around 10, which seems minute?
  - As you progress through the BO procedure and gain more information about the correlations between hyperparameter configurations, it is not really possible to leverage this knowledge to consider novel hyperparameter settings to evaluate meaning you're simply stuck with your initial pool. Do you directly compare against methods with this limitation imposed or not?
- Please clarify the question regarding "normalized regret of utility" (Eq. 3) raised above
- Please clarify the question regarding the horizontal axis raised above
- The ablation study concerning the use of mix-up is interesting (Fig 6). However, have you carried out an analysis to compare the standalone extrapolation performance of your proposed in-context LC curve prediction set-up (with/without mix-up) against more traditional approaches (that may be both simpler and cheaper)?
- *line 406* "Each column corresponds to the cherry-picked examples from each benchmark" - "cherry-picked" is a pejorative term used to describe the practice of isolating only the results that place your method in a good light and/or your competitor's in a bad light. Are the chosen results actually representative of all the other results you could have shown, or are they in fact cherry-picked? If so, we cannot rely upon these reported results to make an assessment of the proposed method!
- *line 446* - how special is the mix-up augmentation strategy? Could you obtain comparable results by fitting an emulator/surrogate model to be able to interpolate entire learning curves between hyperparameter configurations? This would also give you infinitely many training examples that would more accurately preserve correlations between the configurations; granted, it's relatively more expensive but still cheap in absolute terms.
- *line 183* ("utility" function) - It is unclear to me at what stage and exactly how you would elicit user preference data. By generating many pairs of performance-cost pairs upfront and having the user choose their preferences before proceeding with the optimization procedure I assume?
- *line 350* - "We select 23 tasks with 4 different hyperparameters based on [*sic*] SyneTune (Salinas et al., 2022) package" -- what does it mean to select tasks based on some package? You adopt the same set of tasks that they consider in their experimental benchmarks?

---

> ### Author Response · Authors · 2024-11-22
> **Response to Reviewer 8PmF (1/3)**
>
> We sincerely appreciate your time and thoughtful feedback on our work. We have carefully considered your comments and questions and have provided detailed responses below.
>
> ---
>
> **[Q1]** The technical quality of the proposed methodology could be improved. In particular, I found many aspects of the approach to be arbitrary and not well-motivated. For instance, for the stopping criterion described starting line 250, the choice of using the BetaCDF with parameters  β,γ>0 and probability p as the probability of improvement (PI), beyond working fine empirically on the benchmark problems considered, seem highly convoluted and totally arbitrary to me.
>
> - Sorry for the confusion. We were not able to fully provide the motivation for them due to the space constraint. To clarify, note that each criterion, **1) regret-based criterion** and **2) PI-based criterion**, has pros and cons. The regret-based criterion can stop the BO when the utility starts to keep decreasing monotonically, but it is not aware of the possibility that the utility will recover from such downward trends and increase again at some future BO steps. The PI-based criterion can predict such possibilities by extrapolating the learning curves, but there is a risk of overestimation of utility such that it cannot properly stop the BO. We found that we can take the best of the two criteria by smoothly mixing between them. We briefly mentioned it in L269-274.
>
> - The specific reason we use BetaCDF with $\beta, \gamma > 0$ is because it allows us to **easily control the shape of mixing** by tuning $\beta$ and $\gamma$, as shown in Figure 3.
>
> - The specific reason we use PI instead of EI is because EI is sensitive to the scale of utility, which differs from task to task. On the other hand, probability is invariant to the scale, allowing us to **use the same threshold** (0.5 in this case) over the various tasks.
>
> ---
>
> **[Q2]** A major concern I have in the empirical evaluation of the proposed method is in the "normalized regret of utility" (Eq. 3), which is the primary metric that is reported. Beyond being quite complicated to compute (evidenced by lines 417-420), it is also not obvious to me that this is the "holy grail" metric we should be aiming for in the first place. Does this metric not differ depending on the surrogate/extrapolation model of choice?
>
> - The normalized regret of utility used for evaluation is **NOT dependent** on any surrogate/extrapolation model of choice.
>
> - As clearly explained in L377-416, $U_{\text{max}}$ and $U_\text{min}$ is simply the maximum and minimum possible utility achievable assuming that we know the entire LC dataset. We can easily compute $U_\text{max}$ and $U_\text{min}$ from the given LC dataset alone, without any prediction model, similarly to $y_\text{max}$ and $y_\text{min}$ for computing the normalized regret in the previous literature.
>
> ---
>
> **[Q3]** Furthermore, I am unclear as to how this metric is even defined for other methods such as Random, BOHB, etc. which don't explicitly model the performance y, and in which it's unclear how the "utility" can be incorporated? I would be interested in seeing a more conventional plot showing the current best performance (or regret) along the vertical axis.
>
> - Again, the definition of the normalized regret of utility does not require any prediction model.
>
> ---
>
> **[Q4]** Another concern is that the reported empirical results all display the BO iteration along the horizontal axis, which is highly misleading in the context of multi-fidelity BO. It seems to me that the notion of a BO step means totally different things in different frameworks. For instance, in BOHB, a BO step signifies training a model with a particular hyperparameter configuration to full completion, but in most cases they are trained for a fraction, e.g. 27/81 epochs, resulting in a fractional BO step (in this example 1/3rd of a BO step). In contrast, under the proposed framework, a BO step is the advancement of a configuration by a single epoch. Therefore, I am doubtful that the results presented show an apples-to-apples comparison.
>
> - Sorry for the confusion. Throughout the whole paper, the x-axis is actually **“the total epochs spent”**, not “the BO steps”.
> - We corrected all the corresponding figures in the main paper.
> - Note that except for the wrong notation, there is no problem in the comparison itself, because we follow precisely the same comparison procedure in DPL [1], and ifBO [2].
>
> ---

---

> ### Author Response · Authors · 2024-11-22
> **Response to Reviewer 8PmF (2/3)**
>
> **[Q5]** Furthermore, in multi-fidelity BO just showing BO steps along the horizontal axis (fractional or otherwise) is not entirely informative either, especially when cost is of interest. In addition to fractional BO steps, I would like to see a plot with the wall-clock time along the horizontal axis.
>
> - Thanks for your suggestion. However, in this work we do not assume that the BO evaluation time varies across the configurations. While it would be interesting to consider non-uniform evaluation time (e.g., QuickTune [3]) and incorporate it into our utility function, **we still believe that the current focus of this work is complete enough to be a single paper**. We will investigate it as a future work as you suggested.
>
> ---
>
> **[Q6]** A significant weakness of this paper is its lack of clarity, particularly in many parts of the paper (describing important technical details) which I found cryptic and difficult to parse. Some examples include 1) lines 215-219 (details on learning curve extrapolation and how its used to compute the "BO performance") 2) lines 417-420 (details on computing bounds on the "utility") - this is indecipherable
>
> - Could you clarify which parts of L208-215 in revision (original version: L215-219) and L410-414 (original version: L417-420) are hard to understand?
>
> ---
>
> **[Q7]** More generally, this manuscript could benefit from more careful copy-editing. Some specific examples (non-exhaustive) of where writing quality could be improved are enumerated in the "Miscellaneous Remarks" section.
>
> - Thanks for pointing them out. We corrected them in the main paper.
>
> ---
>
> **[Q8]** line 164 - The proposed method works with a fixed, finite pool of hyperparameters. Firstly, I would contend with the claim that this is the "convention" in BO, where it is arguable the exception rather than the rule.
> - The pool-based HPO assumption simplifies the problem to align with widely used tabular benchmark datasets (e.g., LCBench, Taskset, PD1). While we acknowledge that the discretized hyperparameter assumption is limiting, this is unarguably a current convention (e.g., DyHPO [4], DPL [1], ifBO [2], and so on). Also, we did **NOT** mention that this is a “rule”.
> - **We emphasize that this issue extends beyond our paper to the entire multi-fidelity HPO community that currently relies on tabular benchmark datasets for evaluation**. To address this fundamentally, the HPO community needs to propose and adopt new non-tabular benchmark datasets for multi-fidelity HPO.
>
> ---
>
> **[Q9]** However, my biggest question is how this pool is populated in the first place? I would guess randomly, which begs the question of how comparisons are carried out against other methods in which this is not a common practice, e.g. BOHB?
>
> -  This pool is provided by the given dataset, e.g., LCBench, TaskSet, PD1.
> - **We follow precisely the same comparison procedure in the numerous previous works, like DyHPO [4], DPL [1], ifBO [2], and so on**.
>
> ---
>
> **[Q10]** Furthermore, details are missing as to how many hyperparameter configurations there are in this pool. Fig 7c hints that this is around 10, which seems minute?
>
> - **We did clearly provide the details on it in L319-348**.
> - Fig 7c. only shows the distribution of top-10 frequently selected hyperparameter configurations as already explained L466-467.
>
> ---
>
> **[Q11]** As you progress through the BO procedure and gain more information about the correlations between hyperparameter configurations, it is not really possible to leverage this knowledge to consider novel hyperparameter settings to evaluate meaning you're simply stuck with your initial pool. Do you directly compare against methods with this limitation imposed or not?
>
> - All the methods share the same pool of hyperparameter configurations.
>
> ---
>
> **[Q12]** The ablation study concerning the use of mix-up is interesting (Fig 6). However, have you carried out an analysis to compare the standalone extrapolation performance of your proposed in-context LC curve prediction set-up (with/without mix-up) against more traditional approaches (that may be both simpler and cheaper)?
> - Could you elaborate what you mean by “more traditional approaches” here?
>
> ---

---

> ### Author Response · Authors · 2024-11-22
> **Response to Reviewer 8PmF (3/3)**
>
> **[Q13]** line 406 "Each column corresponds to the cherry-picked examples from each benchmark" - "cherry-picked" is a pejorative term used to describe the practice of isolating only the results that place your method in a good light and/or your competitor's in a bad light. Are the chosen results actually representative of all the other results you could have shown, or are they in fact cherry-picked? If so, we cannot rely upon these reported results to make an assessment of the proposed method!
>
> - This is **highly misleading**. If not cherry-picked, then which figure should we present in Figure 5, given only a limited space?
> - As have been done in vast amounts of other works in numerous research areas, such a qualitative analysis is **NOT** for summarizing all the results. Usually, such figures are used for showing how each method behaves/predicts in an intuitive manner. It is Table 1 that shows the average results over all the examples and tasks, not Figure 5.
> - Furthermore, we have **NEVER** mentioned that the readers can rely upon Figure 5 alone. This is why we explicitly mentioned that we cherry-picked those examples, and also provided all the figures in Appendix H.
>
> ---
>
> **[Q14]** line 446 - how special is the mix-up augmentation strategy? Could you obtain comparable results by fitting an emulator/surrogate model to be able to interpolate entire learning curves between hyperparameter configurations? This would also give you infinitely many training examples that would more accurately preserve correlations between the configurations; granted, it's relatively more expensive but still cheap in absolute terms.
>
> - We have done precisely what you described here. Please read our paper more carefully.
>
> ---
>
> **[Q15]** line 183 ("utility" function) - It is unclear to me at what stage and exactly how you would elicit user preference data. By generating many pairs of performance-cost pairs upfront and having the user choose their preferences before proceeding with the optimization procedure I assume?
>
> - Yes, the procedure you assume is exactly the process we described in the paper.
> - Specifically, we assume that users have their own preferences and present them with many performance-cost pairs upfront, allowing them to select their preferred trade-offs.
> - Based on the provided preference data, we fit the utility function and then run our algorithm guided by this utility function.
>
> ---
>
> **[Q16]** line 350 - "We select 23 tasks with 4 different hyperparameters based on [sic] SyneTune (Salinas et al., 2022) package" -- what does it mean to select tasks based on some package? You adopt the same set of tasks that they consider in their experimental benchmarks?
> - Yes. As you mentioned, we adopt the same set of tasks they consider.
> - To clarify, we have revised the wording from "select" to "use" in L344.
>
> ---
>
> **Reference**
>
> [1] Kadra, Arlind, et al. "Scaling laws for hyperparameter optimization." Advances in Neural Information Processing Systems 36 (2023).
>
> [2] Rakotoarison, Herilalaina, et al. "In-Context Freeze-Thaw Bayesian Optimization for Hyperparameter Optimization." arXiv preprint arXiv:2404.16795 (2024).
>
> [3] Arango, Sebastian Pineda, et al. "Quick-tune: Quickly learning which pretrained model to finetune and how." arXiv preprint arXiv:2306.03828 (2023).
>
> [4] Wistuba, Martin, Arlind Kadra, and Josif Grabocka. "Supervising the multi-fidelity race of hyperparameter configurations." Advances in Neural Information Processing Systems 35 (2022): 13470-13484.
>
> ---

---

> > ### Author Response · Authors · 2024-11-25
> > **Kind Reminder**
> >
> > Thank you for your dedication and interest in our paper. As the author and reviewer discussion period approaches its end, we are curious to know your thoughts on our rebuttal and whether you have any additional questions.

---

> ### Comment · Reviewer_8PmF · 2024-11-27
>
> Thanks for providing a detailed response. Some minor points of discussion:
> - Q1-4. Thanks for the clarification.
> - Q6. Please do consider re-writing the paragraphs spanned by specified lines. They are not as clear as you might think
> - Q8. Apologies, "the exception rather than the rule" is used as a [noun phrase](https://www.merriam-webster.com/dictionary/the%20exception%20rather%20than%20the%20rule). When you say it is "unarguably a current convention", I totally agree that this is what this particular thread of works that you cite "(DyHPO [4], DPL [1], ifBO [2], and so on)" adhere to, but I am merely underscoring for the sake of discussion that having a discrete, fixed pool of configurations is a departure from what is traditionally done in Bayesian optimization which is a much broader community than "DyHPO [4], DPL [1], ifBO [2], and so on"
> - Q12. e.g. a simple state-space model
> - Q13. I think a simple fix here is not to use the term "cherry-pick"
> - Q14. You have used a linearly interpolating mix-up strategy and not what has been described, but this is an unimportant point anyway
> - Q15. Thanks for confirming -- this is not as clear as you think.
>
> Overall, my concerns about quality and soundness remain. In particular, I'm still unsure how well-motivated and generally applicable the stopping criterion is, and how beneficial it is to collapse cost and performance into a single value in the generalized manner in which it has been proposed. Additionally, I am still not certain that the empirical results represent a fair apples-to-apples comparison. I understand from the authors' response that the "normalized regret of utility" is well-defined for all methods, but by the same token, the objective function's value (or simple regret) is also well-defined, but much easier to contextualize, and would have gone a long way to address multiple reviewers' concerns. Finally, while I applaud the efforts to compare against the bleeding-edge advanced methods from this year and last, such as DPL and ifBO, I also echo another reviewer's general sentiment that there is a lack of comparison against more established approaches including but not limited to MFKG, MF-UCB, etc.

---

### Author Response · Authors · 2024-11-22
**General Response**

We express our sincere gratitude to all reviewers for their constructive comments and feedback.

We particularly appreciate their acknowledgements on the **clear motivation** (TsBW, Pvea, fTkx), **novelty** (8PmF, wRPD, TsBW), **well-developed** (Pvea, fTkx),  and **impressive results** (wRPD, TsBW, Pvea, fTkx).

---

We have responded to the individual comments from the reviewers below and believe that we have successfully responded to most of them. Here, we briefly summarize the revision of our draft (denoted by blue) requested by reviewers:


- As a response to **8PmF**, we have corrected "BO step" by "Total Epochs Spent" in all figures.
- As responses to **8PmF**, we have corrected typos (L39, 77, 78, 98, 344, 345, 357, 431, and 539).
- As a response to **TsBW**, we have included experiments on the estimated utility function in the first column of Table 2.
- As a response to **TsBW**, we have included a formal definition of multi-fidelity HPO in L161.
- As a response to **TsBW**, we have removed the confusing expression in L374.
- As a response to **fTkx**, we have corrected L55, 64, and 416.
- As a response to **Pvea**, we have included the discussion about MFKG, CFKG, BOCA, MF-UCB in Appendix A.
- As a response to **Pvea**, we have included ablation studies on the proposed stopping criterion, - acquisition function, and transfer learning in Table 3.

---

Please let us know if you have any additional questions or suggestions.

---

### Meta-Review · Area_Chair_RmDf · 2024-12-31

**Metareview:**

This work proposes a cost-sensitive multi-fidelity optimization algorithm that includes a novel transfer learning model based on learning-curve prior fitted networks (LC-PFNs), a preference-based utility model for understanding decision-maker’s preferences with respect to cost and evaluation performance, and an acquisition function.

Reviewers found the combination of transfer learning, cost sensitivity search, and multi-fidelity modeling to be timely and interesting.  The use of mixup for transfer learning with PFNs also appears to be novel and is surprisingly effective.

There were three main issues raised by the reviewers.  First, the motivation for using preference learning is unclear. In particular, why would a human decision-maker be able to more optimally decide when a task should be terminated, compared with a principled algorithm that is able to e.g., compute the information gain from continuing to evaluate (vs starting a new evaluation?) (Pvea, 8PmF).  Second, the acquisition function is heuristic and design choices are not well motivated (TsBW, 8PmF).  Finally, reviewers found the evaluation criteria, such as the use of “normalized regret”, and reporting results in terms of “utility”, which had been criticized as not being inherently meaningful (8PmF, wRPD), nor something that is well-defined or targeted by “baseline” methods (TsBW).

Reviewers have left detailed comments on gaps in the presentation, along with substantive concerns.   The work has a number of moving parts and novel contributions (e.g., the transfer learning, the preference learning, the acquisition function), however, it is not clear why all of these must be combined as they are,  whether all aspects are necessary, and why certain design choices of hyper parameters were selected  (TsBW). Considering each design choice in more detail, and understanding the contribution of each component (while considering more standard MF formulations, such as linear costs, see e.g., comments by Pvea) could help improve understanding here.

This work has a number of interesting ideas and I look forward to seeing future iterations of this work that take into account the thoughtful feedback provided by the reviewers.

**Additional Comments On Reviewer Discussion:**

Reviewers provided detailed feedback which were not adequately addressed by the authors (see MR for detail).

---

### Decision · Program_Chairs · 2025-01-22

Reject